

# High resolution mapping of the NO₂ spatial distribution over Belgian urban areas based on airborne APEX remote sensing

Frederik Tack[1], Alexis Merlaud[1], Marian-Daniel Iordache[2], Thomas Danckaert[1], Huan Yu[1], Caroline Fayt[1], Koen Meuleman[2], Felix Deutsch[2], Frans Fierens[3], and Michel Van Roozendael[1]

[1]BIRA-IASB, Royal Belgian Institute for Space Aeronomy, Brussels, 1180, Belgium
[2]VITO, Flemish Institute for Technological Research, Mol, 2400, Belgium
[3]IRCEL-CELINE, Belgian Interregional Environment Agency, Brussels, 1210, Belgium

*Correspondence to:* Frederik Tack (frederik.tack@aeronomie.be)

**Abstract.** We present retrieval results of tropospheric nitrogen dioxide ($NO_2$) vertical column densities (VCDs), mapped at high spatial resolution over three Belgian cities, based on the DOAS analysis of Airborne Prism EXperiment (APEX) observations. APEX, developed by a Swiss-Belgian consortium on behalf of ESA (European Space Agency), is a pushbroom hyperspectral imager characterised by a high spatial resolution and high spectral performance. APEX data have been acquired under clear sky conditions over the three largest and most heavily polluted Belgian cities, i.e. Brussels, Antwerp and Liège on 15 April and 30 June 2015. Additionally, a number of background sites have been covered for the reference spectra. The APEX instrument was mounted in a Dornier DO-228 airplane, operated by Deutsches Zentrum für Luft- und Raumfahrt (DLR). $NO_2$ VCDs were retrieved from spatially aggregated radiance spectra allowing to resolve urban plumes at the resolution of 60x80 m². Main sources in the Antwerp area appear to be related to (petro)chemical industry while traffic-related emissions dominate in Brussels. The $NO_2$ levels observed in Antwerp range between 3 and 35 x $10^{15}$ molec cm$^{-2}$, with a mean VCD of 17.4 ± 3.7 x $10^{15}$ molec cm$^{-2}$. In the Brussels area, smaller levels are found, ranging between 1 and 20 x $10^{15}$ molec cm$^{-2}$ and a mean VCD of 7.7 ± 2.1 x $10^{15}$ molec cm$^{-2}$. The overall error on the retrieved $NO_2$ VCDs is on average 21% and 28% for the Antwerp and Brussels data set, respectively. Low VCD retrievals are mainly limited by noise (1-sigma slant error), while high retrievals are mainly limited by systematic errors. Compared to coincident car mobile-DOAS measurements performed in Antwerp and Brussels, both data sets are in good agreement with correlation coefficients around 0.85 and slopes close to unity. APEX retrievals tend to be on average 12% and 6% higher for Antwerp and Brussels, respectively. Results demonstrate that the $NO_2$ distribution in an urban environment, and its fine scale variability, can be mapped accurately with high spatial resolution and in a relatively short time frame, and the contributing emission sources can be resolved. High resolution quantitative information about the atmospheric $NO_2$ horizontal variability is currently rare, but can be very valuable for (air quality) studies at the urban scale.



# 1   Introduction

Nitrogen dioxide (NO$_2$) is an atmospheric trace gas and a key pollutant that attracts considerable attention due to several reasons: (1) NO$_2$ is a proxy for air pollution in general, as its abundance mostly coincides with a range of other pollutants; (2) recent studies by the World Health Organization (WHO, 2013) have shown that NO$_2$ exposure can have a direct health

impact; (3) it is a precursor in the formation of aerosols (Chan et al., 2010) and tropospheric ozone (Crutzen, 1970) and therefore it contributes locally to radiative forcing (Solomon et al., 1999), through which it indirectly affects the climate system. In an urban environment, NO$_2$ mainly originates from anthropogenic sources such as the burning of fossil fuels, related to industrial activities and traffic. NO$_2$ is a pollutant with a strong local character and exhibits concentrations that are highly variable in space (horizontal and vertical) and time, often exceeding limits set by the European legislation, i.e. hourly

average limit of 200 µg/m³ and an annual mean limit value of 40 µg/m³ (EU Directive 2008/50/EC). For the reasons stated, the accurate monitoring and mapping of the NO$_2$ variability at high spatial resolution is of great relevance.

In this paper, a method is presented to retrieve tropospheric NO$_2$ vertical column densities (VCDs) from hyperspectral Airborne Prism EXperiment (APEX) observations. A major objective of the study is to assess the technical and operational capabilities of APEX to map the NO$_2$ field at city scale and at high spatial resolution, in a relatively short time and cost-

effective way, and furthermore to characterise all aspects of the retrieval approach. APEX, developed by a Swiss-Belgian consortium on behalf of ESA (European Space Agency), is a pushbroom hyperspectral imager, integrating spectroscopy and two-dimensional (2D) spatial mapping in one single system. The well-established differential optical absorption spectroscopy technique (DOAS; Platt and Stutz, 2008) is applied to the observed backscattered solar radiation, in order to quantify the abundance of NO$_2$ in the atmosphere, based on its fine molecular absorption structures.

For atmospheric NO$_2$ detection, a high spectral resolution is necessary in order to resolve the fast varying spectral signatures of this molecule. High spatial resolution, i.e. higher than 100 m, is also required to identify small scale gradients in the NO$_2$ amounts and to resolve individual emission sources. Although the APEX instrument has been primarily designed for environmental remote sensing of the land surface, it also allows observing atmospheric trace gases as demonstrated in the precursor study of Popp et al. (2012) on APEX tropospheric NO$_2$ mapping over Zurich, Switzerland. In the framework of this

study, dedicated APEX flights were performed above 3 of the largest and most heavily polluted urban areas in Belgium, namely Antwerp, Brussels and Liège. We report on the NO$_2$ retrieval scheme developed at BIRA-IASB and on the results of the flight campaigns performed in Belgium.

For about two decades, systematic mapping of the horizontal distribution of tropospheric trace gases, such as NO$_2$, has been performed at global scale using spaceborne sensors like SCIAMACHY (Scanning Imaging Absorption Chartography),

GOME (Global Ozone Monitoring Experiment), GOME-2, and OMI (Ozone Monitoring Experiment) (see e.g. Richter and Burrows, 2002; Beirle et al., 2010; Boersma et al., 2011; Hilboll et al., 2011; Valks et al., 2011; Bucsela et al., 2013). Such observations are well suited for global monitoring, however, the coarse spatial resolution of typically several tens of



kilometers make them inadequate to detect city-scale $NO_2$ variability and to resolve individual emission sources (see Fig. 1 and 2). Several studies discuss instruments and experiments for atmospheric trace gas retrieval from airborne platforms. Most of these works focus on the retrieval of the vertical distribution of trace gases, based on along-track Multi-AXis (MAX-) DOAS observations (see e.g. Petritoli et al., 2002; Melamed et al., 2003; Wang et al., 2005; Bruns et al., 2006; Dix et al., 2009; Merlaud et al., 2012; Baidar et al., 2013). Only a few recent studies report on the high resolution 2D spatial mapping of the $NO_2$ horizontal distribution from an airborne platform. The discussed hyperspectral imaging systems are based on a whiskbroom (Merlaud et al., 2013b; Liu et al., 2015) or pushbroom setup (Heue et al., 2008; Popp et al., 2012; General et al., 2014; Lawrence et al., 2015; Schönhardt et al., 2015; Meier et al., 2016; Nowlan et al., 2016).

Besides the increased characterisation of horizontal trace gas distribution and related chemical processes in urban areas, high resolution $NO_2$ maps can be valuable for the calibration and validation of satellite products, chemical transport models (CTM) and high resolution air quality models, such as e.g. the RIO-IFDM high resolution air quality model (Lefebvre et al., 2013), developed by the Belgian Interregional Environment Agency (IRCEL-CELINE) in cooperation with the Flemish Institute for Technological Research (VITO). There is also a general growing interest from the remote sensing community for airborne hyperspectral imagers as they can complement and link ground-based measurements, spaceborne observations, and model data.

The paper is organised as follows: Section 2 introduces the optical characteristics of the APEX instrument, as well as the hyperspectral data sets acquired at 3 major Belgian cities. Section 3 discusses post-flight, pre-processing steps in order to improve the signal-to-noise ratio and the spectral performance. Section 4 characterises the key steps of the developed methodology to derive $NO_2$ VCDs from APEX spectra and to produce high resolution $NO_2$ maps, with proper error characterisation. The following section presents and discusses the retrieved $NO_2$ field above Antwerp and Brussels. In Section 6, the retrieved VCDs are quantitatively assessed by intercomparison with correlative data sets (Mobile-DOAS and mini-MAX-DOAS).

## 2  APEX instrument and data sets

### 2.1 APEX optical unit and spatial performance

The APEX instrument was designed and developed by a Swiss-Belgian consortium on behalf of ESA. Since 2011, APEX has been fully operational for data acquisition flights. A brief overview of the optical unit, the core element of the instrument, is given here. Solar radiation, backscattered by the Earth's surface or atmosphere within the field of view (FOV) of the instrument enters the optical unit through a curved slit. A collimator groups and redirects the light towards a beamsplitter, separating the VNIR (Visible Near Infrared; 370 - 970 nm) from the SWIR (Shortwave Infrared; 950 - 2500 nm) channels. The VNIR radiance is spectrally dispersed by a prism, resulting in a greater light throughput and the absence





of overlapping orders when compared to a diffraction grating, but with the drawback of a limited resolution and a non-linear wavelength scale (Platt and Stutz, 2008). The dispersed radiation is then projected on a two-dimensional CCD (Charged Coupled Device) 14 bit depth area detector, recording the intensity in a series of narrow, contiguous spectral bands. The VNIR CCD can record up to 335 unbinned spectral bands. The SWIR channel is not further taken into account in this study

as the DOAS analysis is applied on a small part of the visible wavelength region, i.e. 470-510 nm. The sealed optical unit is enclosed by a thermo-regulated box in order to be temperature (19° C ± 1° C) and pressure (dry nitrogen atmosphere with partial differential pressure control) stabilised. To obtain a good compensation of the aircraft movement, the whole instrument is mounted and operated on a Leica PAV-30 stabilised platform. A more complete description of the APEX instrument, its optical unit and its calibration concept can be found in Itten et al. (2008), Jehle et al. (2010), D'Odorico

(2012) and Schaepman et al. (2015).

The spatial performance and relevant specifications of the APEX sensor are given in Table 1. The pushbroom imaging spectrometer consists of 1000 detector pixels across-track (spatial dimension $x$), which are illuminated simultaneously, and 335 pixels in the spectral dimension $I(\lambda)$. The plane formed by the spatial dimension $x$ and spectral dimension $I(\lambda)$ is called a frame. Mapping of the $NO_2$ distribution below the sensor is performed by the swath imaging of the pushbroom scanner

(spatial dimension $x$) and the forward motion of the airborne platform (spatial dimension $y$). The resulting 3D hyperspectral image cubes, built by acquiring consecutive frames, consist of 2 planimetric dimensions ($x, y$) and a third spectral dimension $I(\lambda)$. With a field of view (FOV) of 28° and a flight altitude of 6.1 km AGL during data acquisition, which is much higher than the sampled air masses containing the bulk of $NO_2$, the swath covers an area of approximately 3 km across-track. The across-track spatial resolution is determined by the instrument's instantaneous field of view (IFOV) and the platform

altitude, while the along-track resolution is determined by the platform motion and the integration time. The ground sample distance (GSD) is 3 (across-track) by 4 (along-track) $m^2$ assuming typical altitude, speed and integration time.

## 2.2 APEX acquisition and data sets

Air pollution levels in the northern part of Belgium are among the highest in Europe. The sources are mainly related to the high population density and related traffic, and regular transport from industrial sources in the Rhine-Ruhr area in Germany.

The annual mean $NO_2$ VCDs for 2015, retrieved based on OMI observations (OMNO2d, Giovanni, NASA GES DISC), range from 1 to 1.2 x $10^{16}$ molec $cm^{-2}$ in the northern part of Belgium and are well above the background levels, as can be observed in Fig. 1. The $NO_2$ levels are around 7 - 8 x $10^{15}$ molec $cm^{-2}$ for the southern part.

The coarse spatial resolution of current global monitoring spaceborne instruments makes them inappropriate for studies of the $NO_2$ field at the scale of cities. This is illustrated in Fig. 2 where the typical spatial extent of a nadir observation from the

OMI and the future TROPOMI (TROPOspheric Monitoring Instrument) spaceborne sensors is illustrated. The latter is the spectrometer payload of the ESA Sentinel-5 Precursor satellite, to be launched in 2017. For the sake of comparison, details of the APEX flightplan over the Antwerp area on 15 April 2015 are provided as well. Whereas the Antwerp city center is





nearly fully covered by one TROPOMI pixel, it is sampled by approximately 5000 APEX pixels, which is needed to detect small-scale $NO_2$ horizontal variability and to resolve individual emission sources. Note that the original APEX pixels of 3 by 4 $m^2$ are spatially binned to pixels of 60 by 80 $m^2$, as will be explained in Sect. 3.1.

The APEX instrument has been installed in the Dornier DO-228 D-CODE, based at the Braunschweig research airport and
operated by Deutsches Zentrum für Luft- und Raumfahrt (DLR). APEX data has been acquired over the 3 largest and most heavily polluted Belgian cities, i.e. Brussels, Antwerp and Liège, on 14, 15 April and 30 June 2015. The 3 acquisition sites are indicated on Fig. 1. Main characteristics of the acquired data sets and covered sites are given in Table 2.

The flight campaign was carefully planned (Vreys et al., 2016a) in order to optimise the acquisition for trace gas retrieval purposes. However, availability of the plane and crew, technical constraints, flight permissions granted by air traffic control
(ATC), and most of all the meteorological conditions seemed to be the main factors constraining and determining the actual flights. APEX was installed in the Dornier DO-228 on 16 March 2015 and starting from that date, a one month window was scheduled to perform the necessary flights. In order to get an optimal $NO_2$ signal, early spring was considered to be a good trade-off: $NO_2$ exhibits a peak in the seasonal cycle around winter in the northern hemisphere. On the other hand, from the perspective of the solar zenith angle (SZA), flights in summer are considered to be more suitable. High sun maximises the
signal or the light backscattered to the sensor and reduces the smoothing of the signal due to shallow sun elevation angle, as discussed in Lawrence et al. (2015). In summer, there are also more clear-sky days. Due to poor weather conditions, actual flights could only be performed at the end of the foreseen time window. The flights were conducted from a small commercial airport, southeast of the Antwerp city center. There were in total 3 flight days. A flight over Brussels was performed on the first campaign day, 14 April 2015. However, due to presence of a substantial amount of scattered clouds,
seriously disrupting the analysis of the data, Brussels was re-flown on 30 June 2015. All other flights were performed under clear-sky conditions and good visibility.

In general, the flightpath consists of adjacent straight flightlines, flown systematically in opposite directions, with overlapping footprints (approximately 20% overlap in the across-track direction between consecutive flightlines), in order to have a gap-free coverage. For example the flightlines above Antwerp and Brussels were alternately flown from south to
north, and from north to south, with the first flightline in the west. The spectra, acquired during the banking of the plane in order to prepare the acquisition of the next adjacent flightline, are not taken into account in the processing, due to the large roll angles. Flights are performed at a constant altitude of 6.1 km AGL, which is well above the planetary boundary layer (PBL) in order to capture the full $NO_2$ tropospheric column. The flights were performed as close as possible to local noon when the sun reaches its highest position, as far this was made possible by ATC and the weather conditions. The 3 cities
were fully covered and also a number of flightlines were extended to rural background sites with decreased levels of $NO_2$. Additionally, a number of background sites were covered when the plane was flying from or to the airport or when flying from one city to another. These reference spectra, containing low $NO_2$ absorption, are used in the DOAS analysis.





## 3    Post-flight pre-processing of APEX data for trace gas retrieval

Hyperspectral observations from the APEX instrument are organised in 3D data cubes per flightline. Different standard APEX products are available: raw level 0 data representing the photon counts on the detector and level 1 data, being the at-sensor radiance after radiometric correction by application of a gain and offset to each detector, based on absolute radiometric calibration with a certified integrating sphere, among others (Hueni et al., 2008). For the purpose of trace gas retrieval in this study a customised product, named level 0-DC, is generated, i.e. a level 0 data product which is spectrally calibrated and corrected for dark current. The dark current is measured at start and end of each acquired flightline, based on a shutter mechanism. This product provides the most suitable spectral performance and the lowest DOAS fit root mean square (RMS) error that can be obtained from the APEX observations.

To reduce the data load and processing time, a spectral subset is taken on the level 0-DC product. The VNIR channel observes radiation in the $370 - 970$ wavelength range while only the subset $370 - 600$ nm is taken into account for the spectral calibration and the DOAS analysis.

### 3.1 Spatial binning and signal enhancement

The raw, spectrally and spatially unbinned APEX spectra, acquired with an integration time of 58 ms, have a typical signal-to-noise ratio (SNR) around 150. In order to reduce the noise level and increase the sensitivity of the instrument to $NO_2$, the raw pixels are spatially aggregated in postprocessing in the along- and across-track direction. If only shot noise is assumed (which is a good assumption for UV-Vis detectors), the noise should decrease with $\sqrt{N}$ according to photon statistics, with N the number of binned spectra.

For a test area, the raw APEX spectra are spatially binned in the along- and across-track direction according to a power of 2, $2^n$, with $n$ ranging from 0 to 8. The number of binned spectra and the corresponding RMS of the noise, from the DOAS fit, are plotted on a double logarithmic scale in Fig. 3. The applied DOAS settings for the noise analysis are provided in Sect. 3.2. The red curve in the Allan plot represents the statistical shot noise $\sqrt{N}$, scaled to the noise of the unbinned raw spectra. High noise, around $7 \times 10^{-3}$, can be observed for the unbinned APEX spectra. The measured noise follows photon statistics well until approximately 200 binned spectra, where a deviation of the noise slope occurs. Due to dominant instrumental noise, such as dark current and read-out noise, and systematic errors in the DOAS fit, saturation occurs and the noise cannot be reduced significantly anymore. The noise is reduced to $4 \times 10^{-4}$ or a favorable SNR of 2500, after binning of 20 pixels in along- and across-track direction. As the curve is slightly deviating from the statistical shot noise $\sqrt{N}$, the improvement of the SNR, compared to the unbinned spectra, is slightly less than 20 times.

The obtained higher SNR improves the detection limit of the instrument at the cost of a reduction of the original spatial resolution. A binning of 20 by 20 pixels is found to be an appropriate trade-off between the obtained instrument sensitivity and the spatial detail, leading to an effective GSD of approximately $60 \times 80$ m$^2$.



The impact of a different spatial binning on the $NO_2$ VCD retrievals is illustrated in Fig. 4. The VCDs retrieved from a co-located column on flightline 8 of the Antwerp data set are plotted from north to south for 3 different binning levels, i.e. $8^2$, $20^2$ and $50^2$ pixels. This results in an effective GSD of 24 x 32, 60 x 80, and 150 x 200 $m^2$, respectively. As can be seen, the 3 $NO_2$ VCD time series show the same patterns of enhanced $NO_2$, consisting of 2 major plumes related to industrial activities

in the Antwerp harbor. However, the $8^2$ binned data contains a lot of noise. On the other hand, the $50^2$ binned data smooths out effective $NO_2$ signals, e.g. at pixel 2050. The smoothing effect can also be observed in Fig. 5. An orthogonal horizontal profile was taken through the second plume and a Gaussian was fitted on the obtained profile $NO_2$ VCDs for the 3 different binning levels. A broadening of the plume can be observed for the higher binning levels. The FWHM increases from 1685, 2129 to 2331 m for $8^2$, $20^2$ and $50^2$ pixels, respectively.

**3.2 APEX spectral performance and wavelength calibration**

A key characteristic of the spectral performance is the instrument spectral response function (ISRF or slit function), being the response of the instrument to a signal as a function of wavelength. The ISRF can be determined by a peak response, i.e central wavelength (CW) and response shape, i.e. full width at half maximum (FWHM). Typically for pushbroom sensors, each across-track detector element should be considered as a 1-D instrument, due to optical aberrations and misalignments,

with an intrinsic spectral response which is slightly different from the others.

In order to determine the spectral response of each pixel and to obtain a good alignment in the DOAS fit between the analysed spectra, reference and the absorption cross-sections, a spectral calibration is performed with the QDOAS software (Danckaert et al., 2015) prior to further processing. The accurate wavelength calibration aligns the Fraunhofer lines in the in-flight APEX spectra with a high resolution solar reference (Chance and Kurucz, 2010). The latter is iteratively convoluted

with an adjusted APEX slit function until a best match with the spectra is found, characterising the effective shift and FWHM. The APEX nominal wavelength-pixel relation, determined under laboratory conditions, is used as initial values in the calibration procedure in order to converge to the best solution. We determine FWHM and shift values in 5 sub-windows between 370 and 600 nm by minimizing the chi-square of the differences between the observed spectrum and the solar reference in each window. Then, we fit polynomials through the 5 resulting shift and FWHM values. As the pushbroom

sensor consists of 1000 pixels across-track, which are binned by 20, the output consists of 50 different calibration sets in total.

During operation, airborne instruments are typically exposed to changes of environmental conditions (changes in pressure, humidity and temperature, vibrations, mechanical stress, etc.) which can affect the instrument characteristics and degrade its performance. Even though the optical unit of APEX is sealed in a nitrogen atmosphere and temperature stabilised, deviations

can be significant between the nominal and effective in-flight spectral performance. This has been extensively investigated in D'Odorico (2012) and confirmed by our study. Nominal parameters of the spectral performance, determined by the laboratory calibration, are given in Table 3. As a prism dispersion element is used, the FWHM is a non-linear function and



broadening with wavelength. The nominal FWHM has a Gaussian shape and is approximately 1.5 nm at 490 nm, i.e. the center of the fitting window, for the nadir detector element. Spectral shift of the CW from an absorption line should be smaller than 0.2 nm according to Schaepman et al. (2015).

The spectral calibration on the in-flight spectra points out a broadening of the ISRF and bigger shifts than specified by the nominal calibration (see Table 3). In Fig. 6, the dependency of the FWHM and the shift on the scanline pixel position are plotted for 490 nm, and subsequently a second order polynomial has been fitted. The spectral calibration output is provided for several flightlines, i.e. 2 flightlines from the Antwerp data set (day 105), 1 flightline from the Liège data set (day 105) and 1 flightline from the Brussels data set (day 181). The FWHM ranges between 2.4 nm, for the outer detectors, and 3.3 nm, for the nadir detector, which is more than twice the nominal value and at the limits to detect subtle spectral variations. Also the spectral shift, exhibiting the typical smile curvature across the detector array (Richter et al., 2011), can be up to 4 times larger than the nominal value, e.g. 0.8 nm in case of the nadir detector on the Brussels flightline 10. Significant changes in FWHM and spectral shift can occur between different flightlines, largely attributed to changing pressure, temperature and environmental conditions, according to D'Odorico and Schaepman (2012) and Schaepman et al. (2015). In Fig. 6, minor changes of the slit function are detected between observations acquired on the same flightline. Bigger changes can, however, be observed between Antwerp flightline 8 and Liège flightline 3, acquired during the same flight (day 105) but with a time interval of approximately 1 hour. Substantial changes of both the FWHM and the spectral shift occur between the Antwerp (day 105) and Brussels (day 181) flight. This time-dependent variability of the APEX slit function in operational conditions can be critical for the analysis of the spectra, as discussed in Sect. 4.2.

Beside the spectral resolution and shift, the spectral sampling interval (SSI) and sampling ratio are important characteristics of the spectral performance. As specified in Table 3, the SSI is approximately 0.9 nm. According to Chance (2005), the Nyquist rate is just met in case of a sampling ratio of 3.1 to 3.6 pixels per FWHM, assuming a Gaussian ISRF. However, being at the limits, APEX is slightly undersampling the spectra, which can complicate the DOAS analysis due to reduced spectral information.

## 4   NO$_2$ vertical column density retrieval algorithm

The flowchart in Fig. 7 illustrates the key steps of the applied NO$_2$ VCD retrieval algorithm. For each of the modules, the main input and output data sets and their respective file formats are specified. A NO$_2$ vertical column can be derived for each APEX pixel or measured spectrum $i$, according to:

$$VCD_i = \frac{SCD_i}{AMF_i} \qquad (1)$$

In Eq. (1), the NO$_2$ $VCD_i$ or the integrated amount of molecules per cm$^2$ expected for a single, vertical transect of the atmosphere is defined as the ratio of the measured slant column density or the number of molecules per cm$^2$ detected in an



observation ($SCD_i$) and a corresponding air mass factor ($AMF_i$) (Solomon et al., 1987). The direct output of the DOAS analysis (see Sect. 4.1) is not $SCD_i$, but a differential slant column density (D$SCD_i$), being the difference of the concentration of $NO_2$ integrated along the effective light path and the $NO_2$ concentration in a reference spectrum ($SCD_{ref}$). The residual $NO_2$ amount in the background spectrum needs to be determined in order to convert D$SCD_i$ to $SCD_i$ (see Sect. 4.2). Therefore Eq. (1) needs to be rewritten as:

$$VCD_i = \frac{DSCD_i + SCDref}{AMF_i} \tag{2}$$

Or

$$VCD_i = \frac{DSCD_i + (VCDref * AMFref)}{AMF_i} \tag{3}$$

AMFs account for enhancements in the optical path length of the slant column, due to viewing and sun geometry, albedo, aerosol scattering and the $NO_2$ vertical profile (see Sect. 4.3).

### 4.1 DOAS analysis of the measured spectra

The obtained binned APEX spectra are analysed in the 470-510 nm visible wavelength region by application of an adapted version of the QDOAS non-linear least-squares spectral fitting tool, developed at BIRA-IASB (Danckaert et al., 2015). The adapted version allows processing the spectra of full APEX flightlines simultaneously, with both the input and output in a network common data form (netCDF) file format. Note that the small 470-510 nm fitting window is not considered to provide the highest sensitivity to $NO_2$. Instruments specifically designed for $NO_2$ trace gas retrieval usually operate in the 425-490 nm interval. This broad window contains more spectral information and strongly structured $NO_2$ absorption features, while there is low interference with absorption structures of other trace gases, providing optimal sensitivity to $NO_2$. However, interference with unidentified instrumental artefacts or features prevents us from extending the fitting window to lower wavelengths than 470 nm. Currently, the chosen wavelength interval is considered to be the best trade-off between sensitivity on the one hand and minimum interference with other absorbers and instrumental structures on the other hand.

The basic idea of the DOAS approach is to separate broadband and narrow band signals in the spectra and to isolate the narrow molecular absorption structures (widths usually smaller than a few nm). Beside relevant high-pass filtered trace gas laboratory cross-sections ($NO_2$ and $O_4$), a synthetic Ring spectrum, a synthetic resolution cross-section and a low-order polynomial term are fitted to the logarithm of the ratio of the observed spectrum, and a reference spectrum. They account for respectively (1) the Ring effect (Grainger and Ring, 1962), i.e. the filling-in of Fraunhofer lines by rotational Raman scattering on air molecules (2) small differences in spectral resolution between the reference and the spectrum to analyze, due to the unstable slit function, and (3) smooth broad-band variations (e.g. reflection at the Earth's surface) and Rayleigh




and Mie scattering. $O_3$ and $H_2O$ cross-sections were not fitted due to cross-correlations and overparameterisation of the small fitting interval. Further details about the main DOAS settings can be found in Table 4.

A typical DOAS fit of an APEX spectrum is illustrated in Fig. 8. The direct output or resulting coefficients of the applied fitting algorithm are $NO_2$ differential slant columns. A $NO_2$ DSCD of $2.61 \times 10^{16}$ molec cm$^{-2}$ is retrieved with an RMS on the

residuals of $4.03 \times 10^{-4}$. In the absence of systematic structures, the RMS is the standard deviation of a Poisson distribution, corresponding to the measured photons. The RMS corresponds with a favorable SNR of 2500, and is in line with the obtained signal enhancement after spatial binning, as discussed in Sect. 3.1.

## 4.2 Background spectrum

In the DOAS analysis, the concentration of $NO_2$ is determined with respect to an unknown amount of $NO_2$ in a selected

reference spectrum ($SCD_{ref}$). This differential approach neutralises both systematic instrumental instabilities and the prominent Fraunhofer lines in the spectra, which blur out the much finer trace gas absorption structures. This approach also cancels out the stratospheric $NO_2$ contribution to the signal, making the measurements only sensitive to tropospheric absorption, under the assumption that the stratospheric $NO_2$ field has a negligible spatial and temporal variability in the time between the acquisition of the reference spectrum and the measurements. At daytime, the stratospheric $NO_2$ column is

characterised by a near-linear slow increase due to the photolysis of $N_2O_5$ at mid-latitudes. The diurnal increase of the $NO_2$ stratospheric column between 80° SZA sunrise and sunset is estimated to be approximately $1 \times 10^{14}$ molec cm$^{-2}$ per hour (Tack et al., 2015) and thus much smaller than the retrieved VCDs. The latter have a mean value of $1.7 \times 10^{16}$ and $7.7 \times 10^{15}$ molec cm$^{-2}$ for the Antwerp and Brussels data set, respectively. All flights were performed close to local noon, and in general there was a minor time interval between the acquisition of the spectra and the reference of less than one hour.

The residual $NO_2$ amount in the reference ($SCD_{ref}$) is an unknown that needs to be estimated, which is a general shortcoming of all airborne DOAS pushbroom imagers. Some studies assume that there is no (Schönhardt et al., 2015) or very little, e.g. $1 \times 10^{15}$ molec cm$^{-2}$ (Popp et al., 2012), tropospheric $NO_2$ in a background area. Due to a combination of the nature of the APEX instrument and the study area, these assumptions were not valid, as explained here:

-The effective slit function is affected by environmental conditions during operation of APEX, despite the fact that the

instrument is sealed, pressure- and temperature-stabilised. Fig. 6 shows that changes of the slit function can be significant for different flightlines within a same flight, and certainly for different flight days. The unstable slit function can cause misregistrations and spurious residuals in the DOAS fit. Even when the latter are small, they can impact the retrievals, considering that very fine absorption structures are analysed. As a consequence, a reference cannot be used to analyse a certain spectrum if the spectral performance deviates too much. Quality parameters of the fit, such as the RMS, were

carefully checked in order to detect significant changes of the effective slit function between the analysed spectra and reference. In case of the Brussels data set (day 181), reference data were selected per individual flightline due to the larger instability affecting the slit function.





-The nature of the reference area further complicates the estimation of $SCD_{ref}$. Based on a priori information, a number of candidate background areas around the cities were covered during the taxi flight between airport and survey area, far from emission sources and with decreased levels of $NO_2$. Due to the unstable slit function and its time-dependency, unfortunately these reference flightlines could not be used in the analysis in most cases. Instead, the references needed to be selected closer to the city, where the $NO_2$ levels are relatively high and generally have a strong spatial variability. Both for the Antwerp (day 105) and Brussels (day 181) data set, the references were selected in the south part of the flightlines, upwind of the main sources in the city. The $SCD_{ref}$ was estimated based on co-located mobile-DOAS measurements. Following Eq.(2), the retrieved VCD will be overestimated in case the effective $SCD_{ref}$ is lower than the estimated amount.

### 4.3 Air mass factor calculations

A slant column depends on multiple light paths of backscattered solar radiation, contributing to the spectrum observed by the instrument. In order to derive the effective optical path length through the atmosphere and thus to be able to interpret and compare observations, transfer of radiation in the atmosphere needs to be modelled and appropriate enhancement factors need to be calculated. In this study, $NO_2$ box-AMFs ($BAMF_i$) have been calculated with the linearised radiative transfer model (RTM) LIDORT 2.6 (Spurr, 2008). The Box-AMFs describe the sensitivity of the measurements as a function of altitude, resulting in a height-dependent assessment of the instruments sensitivity (Wagner et al., 2007). The radiative transfer equation is solved in a multi-layer, multiple scattering atmosphere using the discrete ordinate method. RTM simulations are performed at 490 nm, i.e. the middle of the $NO_2$ fitting window. 93 atmospheric height layers $j$ are defined from the ground surface to the top of the atmosphere (TOA), the latter determined at 120 km altitude, and for each layer a box-AMF is retrieved. The vertical discretisation consists of 40 layers of 50 m thickness, until 2 km and 12 layers of 500 m between 2 and 8 km. Above 8 km, the altitude grid of the US Standard Atmosphere is adopted. A total AMF ($TAMF_i$) can be derived for each APEX spectrum $i$ by integration of the $BAMF_j$ along an a priori $NO_2$ vertical profile:

$$TAMF_i = \frac{\sum_{j=0}^{TOA} BAMF_j * VCD_j}{\sum_{j=0}^{TOA} VCD_j} \tag{4}$$

In Eq. (4), $BAMF_j$ and $VCD_j$ refer to the box-AMF and the a priori partial $NO_2$ VCD of atmospheric layer $j$. A mean TAMF of 1.9 is obtained for the Antwerp and Brussels data set.

### 4.3.1 RTM parameters

LIDORT numerically reproduces the state of the atmosphere and transfer of the solar radiation through the atmosphere based on a priori information on all parameters that affect the lightpath, e.g. the surface albedo, sun and observation geometry, and atmospheric properties (pressure, temperature, cloud cover, absorber and aerosol vertical profiles). (1) The surface albedo, as well as (2) the sun and viewing geometry, i.e. SZA, viewing zenith angle (VZA) and relative azimuth angle (RAA), can be



extracted from the observations. These additional (meta)data sets are provided for each observed spectrum and are spatially binned accordingly. The surface albedo is approximated by the APEX reflectance value (Level 2 product), which is obtained for each pixel after application of an atmospheric correction algorithm with MODTRAN4 (Berk et al., 1999) on the radiometrically calibrated level 1 at-sensor radiance product (Biesemans et al. 2007, Sterckx et al. 2016). The viewing and sun geometry are output products of the APEX orthorectification module, as described in Sect. 4.4. (3) Pressure and temperature profiles are provided by the US Standard Atmosphere at mid-latitude. (4) Clouds usually introduce a major uncertainty in atmospheric modelling. However, all flights were performed under clear-sky conditions avoiding the necessity of a cloud retrieval scheme. (5) A priori $NO_2$ vertical profile shapes are obtained from the regional-scale air quality model AURORA (Air quality modelling in Urban Regions using an Optimal Resolution Approach; Lauwaet et al., 2014) at a high resolution of 1 x 1 $km^2$. The AURORA model consists of a fine vertical grid with 40 layers of 50 m thickness and 37 layers of 100 m thickness to a maximum altitude of 5700 m. Model output for the time of flight and overpass location was computed. Interpolation on the coupled 3D model grid provides a local $NO_2$ vertical profile for each APEX pixel. The obtained $NO_2$ profile shapes are assessed in the next section. (6) Aerosols can both enhance or reduce the AMF, depending on their vertical distribution. Since the latter information was not available and since all flights took place on clear spring/summer days with good visibility, a pure Rayleigh atmosphere was considered to compute the VCDs. The uncertainty related to this assumption is, however, discussed and quantified in the next section.

### 4.3.2 RTM sensitivity study

To study the impact of the input parameters on the AMF computations, sensitivity tests were performed based on different scenarios with varying input in the radiative transfer modelling. First, the mean and 1-sigma standard deviation were calculated for the albedo, RAA, VZA and SZA based on 73000 pixels/observations of the Antwerp data set. In Table 5, each row corresponds to an RTM parameter for which 2 scenarios are provided: mean +/- 1-sigma level (68%). For the study of a certain parameter, the other RTM parameters are assigned a fixed value μ, being 5%, 94.1°, 7°, and 54.6° for the albedo, RAA, VZA and SZA, respectively. For each scenario, corresponding TAMFs are computed.

For the sun and viewing geometries, the TAMF variability is low, i.e. less than 6% for input within μ ± 1σ. From the studied input parameters, the surface albedo has clearly the most significant impact on the AMF computations, which is consistent with previous investigations such as Boersma et al. (2004) and Lawrence et al. (2015). The TAMF variability can be up to 65% for albedo input within μ ± 1σ, illustrating the importance of having accurate knowledge of the surface properties in case of airborne imaging applications. The strong dependency of the TAMF computations to the albedo is illustrated in Fig. 9. Similar statistics were obtained for the Brussels data set.

In order to obtain a better understanding of the instrument vertical sensitivity and the impact of the albedo on the radiative transfer, BAMFs are plotted as a function of the altitude in Fig. 10 for 5 different surface albedo scenarios. The sensitivity of the instrument is strongly height-dependent. The sensor is mostly sensitive to $NO_2$ observed directly below the airplane, with



its sensitivity decreasing towards the ground surface, except for the scenario with very high albedo (45%). In case of high albedo, much of the incident radiance is reflected towards the sensor, increasing its sensitivity to $NO_2$ and thus significantly impacting the compensation factor. The sensitivity is about 2.9 directly under the plane and decreases to 0.3 to 2.1 at the ground surface for an albedo of 0.3 to 8%, respectively. The sensitivity to the atmospheric layers above the sensor is almost

constant and relatively low, around 1.7. This is very close to the geometrical AMF (1/cos(SZA)).

While the $NO_2$ horizontal distribution can be mapped, based on airborne APEX hyperspectral data, the details of the vertical distribution of $NO_2$ in the atmosphere are not well known. As discussed in Sect. 4.3.1, in this work, a time and space dependent $NO_2$ profile is interpolated on the high resolution AURORA 3D model grid for each APEX pixel. Then, TAMFs are computed based on integration along the obtained a priori $NO_2$ profile. In Figure 11, the mean AURORA profile is

plotted for the Antwerp study area. 3 extreme cases of local AURORA profiles are plotted as well, i.e. for an urban, industrial and semi-rural site. Over the harbor, with many industrial activities, large $NO_2$ concentrations can be observed for the surface layer. Also the higher atmospheric layers contain relatively more $NO_2$, probably due to emitting stacks which can be as high as 70 m. Over the city center, the bulk of the $NO_2$ can be related to traffic emissions at the surface.

The impact of the $NO_2$ profile shape on the TAMF computation is reported in table 5 by comparison of 2 scenarios:

integration of the BAMFs along (1) local $NO_2$ vertical profiles $A_{interp}$, interpolated on the AURORA model grid and (2) a fixed AURORA $NO_2$ profile $A_{harbor}$, with very high $NO_2$ concentrations for the surface layers, and which can be assumed to be an extreme case. A scatter plot and linear regression analysis for the TAMFs, obtained by both scenarios, are shown in Fig. 12. The higher surface concentrations of the $A_{harbor}$ profile give a larger weight to the relatively low BAMFs at the surface (see Fig. 10) resulting in a mean TAMF decrease of 7.5%, when compared to the first scenario. This impact reduces

to zero at higher TAMF values, which can be related to higher albedo values, which have a much smaller decrease in sensitivity to the surface. In a second sensitivity test, the TAMFs obtained by integration along local AURORA $NO_2$ vertical profiles $A_{interp}$ are compared with TAMFs computed based on a simple $NO_2$ box profile of 0.5 and 1 km height and well-mixed in the boundary layer. In case of a $NO_2$ box profile of 0.5 km, TAMFs are on average 7% smaller when compared to local AURORA $NO_2$ profiles, while TAMFs are slightly higher (~1%) in case of a box profile of 1 km.

Previous studies (Leitao et al., 2010; Meier et al., 2016) indicate that aerosols can enhance or reduce the AMF, depending on their position with respect to the $NO_2$ layer, the optical thickness, and the absorption of the aerosol layer. An aerosol optical thickness (AOT) lower than 0.15, at 500 nm, was measured during all flights by a CIMEL sun photometer at the AERONET station (Holben et al., 1998) in Uccle (50.78° N, 4.35° E, 100 m ASL). Radiative transfer simulations with a corresponding well-mixed extinction at the surface yield an albedo-dependent aerosol effect of 10% or less when compared to the AMFs

computed based on a Rayleigh atmosphere. This relative uncertainty is considered for all flights.




## 4.4 Postprocessing: destriping

After application of the retrieval equation, i.e. Eq. (3), a bias correction is applied on the retrieved $NO_2$ VCDs to cope with the across-track stripe-like pattern in the generated maps. Striping is inherent to pushbroom sensors due to the intrinsic spectral response of each detector which is slightly different from the others (see Sect.3.2). The applied bias correction is

based on the algorithms presented in Boersma et al. (2011), Popp et al. (2012) and Lawrence et al. (2015). For each flightline, the column $NO_2$ values are averaged and a third order polynomial is fitted to the column averages. The deviation from the polynomial is treated as a detector dependent bias and used as a correction factor to subtract from the retrieved $NO_2$ columns. The procedure removes the across-track striping to a great extent, while retaining the $NO_2$ spatial patterns. For visualisation purposes, the retrieved VCDs are smoothed by convolving a fifth order Savitzky-Golay least-squares

polynomial filter (Savitzky and Golay, 1964; Schafer, 2011) in the across-track direction.

## 4.5 Spectra geolocation and georeferencing

The APEX sensor is equipped with a high-grade Applanix POS/AV 410 navigation system, which records sensor position, i.e. latitude, longitude and elevation, and orientation, i.e pitch, roll and heading. Concurrently, global positioning system (GPS) base station data, for differential correction, and data originating from an inertial measurement unit (IMU) are

recorded. All telemetry is blended in real-time and logged for post-processing to allow proper georeferencing of the spectra (Mostafa and Hutton, 2001). On 14 April also a boresight calibration flight (Mostafa, 2001) was performed over Oostend, Belgium for accurate georeferencing purposes. The boresight angles account for misalignments between the IMU axis and the sensor axis and they are computed every time APEX is mounted in the aircraft. Ground control points (GCPs) selected from orthophotos are identified in the APEX data and, following a Monte Carlo procedure, optimal parameters to

compensate for roll, pitch and yaw errors are inferred, greatly reducing geolocation errors which are usually lower than one unbinned spatial pixel (Vreys et al., 2016b). The orthorectification and georeferencing module receives the sensor interior and exterior orientation, boresight calibration data and digital elevation model (DEM) data as input. The module outputs the position and the complete viewing geometry for each pixel or measured spectrum, allowing a proper mapping of the retrieved $NO_2$ spatial distribution. In Vreys et al. (2016b), the georeferencing module and its qualitative and quantitative

assessment is discussed more deeply.

The georeferenced VCDs and intermediate products, e.g. DSCDs and AMFs, are eventually gridded and overlayed on Google Maps layers in an open-source geographic information system (GIS) environment, QGIS 2.10.1 (QGIS development team, 2009).

## 4.6 $NO_2$ VCD error budget

The overall error on the retrieved $NO_2$ VCDs originates from uncertainties in the calculated DSCDs, $SCD_{ref}$, and AMFs. One assumes that the contributing uncertainties are sufficiently uncorrelated as they arise from nearly independent steps. Based





on Eq. (2), the overall error of the NO$_2$ VCD retrieval algorithm can be quantified based on the following error propagation method:

$$\sigma^2_{VCD_i} = \left(\frac{\sigma_{DSCD_i}}{AMF_i}\right)^2 + \left(\frac{\sigma_{SCD_{ref}}}{AMF_i}\right)^2 + \left(\frac{SCD_i}{AMF_i{}^2} \times \sigma_{AMF_i}\right)^2 \qquad (5)$$

(i) The random error on the DOAS fit (1-sigma standard deviation), $\sigma_{DSCD_i}$, is a direct output of QDOAS for each fit and has a typical value between 3.4 and 4.4 x 10$^{15}$ molec cm$^{-2}$ on the APEX DSCD retrievals, when spectra are binned 20 by 20 pixels. Note that this is approximately one order of magnitude higher than for the fixed ground-based stations, e.g. the BIRA-IASB MAX-DOAS instrument (Tack et al., 2015), having a higher SNR and better spectral performance. Whereas $\sigma_{DSCD_i}$ is a rather minor error source in case of ground-based stations, it becomes a significant contributor to the total error in case of an airborne imager.

(ii) The second error source originates from the estimation of the NO$_2$ residual amount in the reference spectra, $\sigma_{SCD_{ref}}$. As SCD$_{ref}$ is determined from co-located mobile-DOAS measurements, the overall error on the mobile-DOAS retrievals is taken into account. The mobile-DOAS error estimation is discussed in Merlaud (2013a) and Constantin et al. (2013). The mean overall error for the Antwerp and Brussels data set is 1.8 x 10$^{15}$ molec cm$^{-2}$.

(iii) The error in the calculation of the air mass factor, $\sigma_{AMF_i}$, is caused by the uncertainties in the assumptions made for the radiative transfer model parameters (See Sect. 4.3.1). The contributing uncertainties can be summed in quadrature to obtain an overall error estimate $\sigma_{AMF_i}$. According to Boersma et al. (2004), the error budget associated with the computation of the AMF is dominated by the cloud fraction, surface albedo, and NO$_2$ profile shape: (1) As flights were performed under clear-sky conditions, cloud fraction is not considered an error source in this case. (2) Sensitivity tests, performed in Sect. 4.3.2, indicate that the surface albedo has the most significant impact on the effective lightpath, thus on the AMF. Within the albedo 1-sigma interval, the AMF variability can be up to 65%. However, as absolute radiances can be directly derived from the APEX instrument, the albedo can be determined with relatively high accuracy. For a realistic estimate of the uncertainty, following study was performed: several albedo types were measured in the field with an ASD FieldSpec 4 spectrometer (http://www.asdi.com/products-and-services/fieldspec-spectroradiometers/fieldspec-4-hi-res, last access: 12-09-2016) and compared to the APEX surface albedo. For the wavelength 490nm, the average albedo error over all targets is 10%, which is assumed to be a realistic estimate of the uncertainty related to the a priori surface albedo. (3) Based on the sensitivity study performed in Sect. 4.3.2, the uncertainty related to the a priori NO$_2$ profile shape is lower than 8%. (4) According to the performed simulations, the uncertainty related to the assumption of a pure Rayleigh atmosphere is estimated to be less than 10%. (5) Both the viewing and sun geometry can be determined with high accuracy, thus the impact on the error in the AMF computation is expected to be small. Moreover, the performed sensitivity study, summarised in table 5, has revealed that





varying input for the viewing/sun geometry has a very low impact on the TAMF variability. Therefore it is assumed that the uncertainties related to RAA, VZA and SZA are less than 1%. Finally, all error sources contributing to the overall error $\sigma_{AMF_i}$ are summed in quadrature and an estimate of approximately 15% is obtained.

The error budget analysis, based on 73000 observations of the Antwerp data set, is reported in Table 6 and Fig. 13. In Table 6, typical relative and absolute errors are given for different classes, depending on the $NO_2$ VCD amount. For each class, the typical error $\sigma$ is provided for each individual error source in the retrieval approach, as well as for the overall error on the retrieved VCDs, $\sigma_{VCD}$.

An overall relative error, $\sigma_{VCD}$, of 21% on the retrieved $NO_2$ VCDs is reported in Table 6 for the Antwerp data set, or a mean VCD and absolute error of $17.4 \pm 3.7 \times 10^{15}$ molec cm$^{-2}$. For the Brussels data set the relative error is slightly higher (~28%), due to a lower mean VCD of $7.7 \pm 2.1 \times 10^{15}$ molec cm$^{-2}$. In general, the main contribution is coming from the error on the DOAS fit (1-sigma slant error), $\sigma_{DSCD}$, which can be quite substantial in case of small VCD retrievals (~29% for the Antwerp data set). For medium and high VCDs, the overall relative errors decrease to 21% and 18%, respectively. For the latter, the main contributions are from the error on the DOAS fit and the TAMF computation, $\sigma_{TAMF}$. Relative errors related to the DOAS fit and to the estimation of the residual amount, $\sigma_{SCDref}$, drop in case of larger VCDs, while errors originating from the TAMF computation are not affected. Of course the overall relative error is dependent on the $NO_2$ levels in the covered area. Low VCD retrievals will produce larger relative errors. The relatively low mean $NO_2$ VCD and related higher error for the Brussels data set is due to a combination of the seasonality of $NO_2$, the lack of significant industrial sources in the area and the fact that a fair amount of semi-rural area around the city was covered (see Sect. 5).

The overall absolute and relative errors are plotted in function of the retrieved $NO_2$ VCDs in Fig. 13, for the Antwerp data set. The largest absolute errors are obviously associated with the highest retrievals. The relative errors, on the other hand, which can be up to 100% in case of very low tropospheric contributions, show a steep and rapid drop in case of increasing VCDs. The relative error is almost constant (~21%) for $NO_2$ VCDs larger than $1.5 \times 10^{16}$ molec cm$^{-2}$.

## 5    Discussion of the retrieval results

The generated $NO_2$ VCD distribution maps are shown in Fig. 14 and 16 for respectively the Antwerp (day 105) and Brussels (day 181) data set. Meteorological conditions and the general flight pattern, both important to interpret the observed $NO_2$ field, are discussed in Sect. 2.2. In general, the obtained spectral and spatial resolution allows to map the fine scale $NO_2$ horizontal variability and strong spatial gradients, and to resolve individual emission sources. Patterns of enhanced $NO_2$ can be observed, largely consistent with and transported downwind from the emission inventory sources. Structures are evidently exposed within the detected plumes. The distribution maps show that the $NO_2$ concentrations can be highly variable in urban areas and can exhibit strong gradients. The maps are built from several adjacent flightlines with an approximated acquisition



time of 8 to 15 min per flightline. As a dynamic $NO_2$ field is measured, minor biases can occur between adjacent flightlines related to transport.

In the case of the Antwerp data set, the predominant anthropogenic emitters are mainly related to industrial activities in the harbor, but also to traffic in the southeast part. The port of Antwerp contains the biggest (petro)chemical cluster in Europe

with branches of BASF, ExxonMobil, Solvay, Total, etc. The red dots in Fig. 14 represent the most significant stacks, emitting more than 25 kg of NOx per hour, according to the emission inventory of the Belgian Interregional Environment Agency. The $NO_2$ field exhibits a strong gradient from west to east, consistent with the southwesterly wind direction. In the west, the $NO_2$ levels are low, around 3 to 7 x $10^{15}$ molec cm$^{-2}$ due to the lack of major contributing sources. Substantial uncertainties can occur in this area as the levels are lower than or close to the detection limit. Downwind of the sources, the

transported $NO_2$ is building up and patterns of enhanced $NO_2$ can be observed with maxima up to 3.5 x $10^{16}$ molec cm$^{-2}$. The $NO_2$ VCDs are on average 1.7 ± 0.4 x $10^{16}$ molec cm$^{-2}$. The detected plumes are clearly related to and transported from the most contributing stacks in the area. The main central plume with large extent is a double plume with main emissions from 2 chimney stacks, emitting respectively 30 and 95 kg of NOx per hour at an altitude of 70 m AGL, and a third stack northeast of it, emitting 145 kg of NOx per hour at an altitude of 35 m AGL. The plume is approximately 12 km long and is

unfortunately not fully covered by the flightplan. There is actually a cluster of 30 to 40 stacks which are contributing as well to the central plume. However, according to the emission inventory the emissions are mostly lower than 10 kg of $NO_x$ per hour. Ship emissions are also expected to contribute to the observed $NO_2$ field, however these are hard to differentiate in this particular data set.

In the southeastern part of the $NO_2$ map (Fig. 14), $NO_2$ patterns can be observed, which are related to traffic emissions from

Antwerp city and R1 ring road. The 2 last flightlines are acquired around 11 - 11u30 AM LT and presumably the air masses containing the emissions from the rush hour are detected here, transported from the city and ring road and building up, due to low wind speeds, northeast of the city. In the western part of the data set, an artefact is still present over the H-shaped docks due to too low retrieved VCDs over the water body and overestimation of the VCDs over the surrounding quays. Such an artefact is not present over other water bodies in the data set. The artefact could not be removed properly by selecting a

larger amount of $NO_2$ in the reference spectrum. It is most probably caused by a combination of the overall low retrievals in this area, upwind of the sources, and computed AMFs that don't properly compensate for the lower backscatter over the water body (low albedo), and higher backscatter over the surrounding docks (high albedo). As discussed in Sect. 4.3.2, the albedo has a strong impact on the AMF computations. The albedo varies strongly from approximately 2% to 12% between the water body and the quay. A slight overestimation of the albedo, and subsequently the AMF, over the waterbody and

underestimation of the albedo and AMF over the quay could explain this artefact.

On flightline 8 of the Antwerp data set, an along-track profile is taken (see Fig. 14) for which the $NO_2$ DSCD and VCD time series are plotted from north to south in Fig. 15. Negative DSCDs point at a high amount of $NO_2$ in the reference/background, when compared to the analysed spectrum. The blue shaded error region on the DSCDs corresponds to





the statistical error on the DOAS fit, being also a measure for the detection limit. In order for an absorber to be clearly identified, the retrieved column needs to be larger than this threshold. The 1-sigma detection limit of APEX retrievals has a typical value between 3.4 and 4.4 x $10^{15}$ molec cm$^{-2}$ on the DSCD, corresponding to 1.8 and 2.3 x $10^{15}$ molec cm$^{-2}$ on the VCD, assuming a typical AMF of 1.9 (See Sect. 4.3). The retrieved $NO_2$ signal is well above the detection limit.

The $NO_2$ VCD distribution map for the Brussels data set is shown in Fig. 16. The study area, consisting of the Brussels (sub)urban area and surrounding background, is covered by 95833 binned APEX pixels in approximately 80 minutes. In general, the $NO_2$ levels are almost 55 % lower than for the Antwerp data set, with minima and maxima of 1 and 20 x $10^{15}$ molec cm$^{-2}$, respectively. This is due to both the lack of significant industrial sources in the Brussels area, as well as the seasonality of $NO_2$ which tend to show maxima in winter and early spring. The $NO_2$ VCDs are on average 7.7 ± 2.1 x $10^{15}$

molec cm$^{-2}$. Again a strong gradient can be observed, consistent with the southeasterly wind direction, with low $NO_2$ VCDs above the Sonian forest in the southeast and increased levels downwind of Brussels city.

The $NO_x$ sources in Brussels are predominantly related to traffic and concentrated along the R0 Brussels ring road and the junctions with the key highways E40 and E19. The R0 is one of the busiest highways in Belgium with traffic volumes of more than 70.000 cars per day. Patterns of enhanced $NO_2$ can be observed near the city center as well. In Fig. 17, a number

of noticeable hotspots of interest in the Brussels data set, indicated by blue, dashed squares in Fig. 16, are highlighted and discussed here:

- Some of the highest emissions can be observed in the northeast where there is a major junction between the ring road R0 and the E19 (Fig. 17.a). Besides, there are 2 other main interchanges and some small industry in Vilvoorde. Strong $NO_x$ emissions in this area can also be related to planes and airport traffic operations at the Brussels international airport. In June

2015, there were in total 22338 plane movements and on 30 June, approximately 50 planes took off in northeastern direction between 3 and 6 PM LT. However due to the proximity of the ring road and interchanges, it is nontrivial to differentiate the contributing sources.

- An increase of the $NO_2$ VCDs can be observed at "place Meiser" with values around 1.2 x $10^{15}$ molec cm$^{-2}$ (Fig. 17.b). It is a busy junction, close to the city center, where 7 major roads are coming together.

- Fig. 17.c zooms in on the eastern part of the E40 highway, just past the junction with the ring road. The highway forms a clear segregation between the $NO_2$ levels in the northern and southern part. Due to the lack of considerable contributing sources, the air masses upwind of the highway contain little $NO_2$, while there is a significant increase noticeable downwind of it.

- The gas turbine Drogenbos powerplant in the southwest is the only significant industrial source, within the covered area,

emitting more than 25 kg of NOx per hour. A confined $NO_2$ emission plume, transported downwind from the double stack close to the channel, can be clearly resolved with a typical VCD around 1.5 x $10^{16}$ molec cm$^{-2}$ (Fig. 17.d).





The NO$_2$ levels observed in Liège range between 1 and 32 x 10$^{15}$ molec cm$^{-2}$, with a mean VCD of 13.3 ± 3.1 x 10$^{15}$ molec cm$^{-2}$. The overall error on the retrieved NO$_2$ VCDs is on average 23%. In order to avoid repetition, the small Liège data set is not shown and further discussed here as it leads to similar findings and conclusions.

## 6  Correlative data sets

### 6.1  mobile-DOAS observations

In order to quantitatively assess the NO$_2$ retrievals, they are compared with correlative data sets acquired by 2 car mobile-DOAS systems. The routes strive to have a good spatial distribution by covering the city ring road, and transects to the city center and other main sources. The BIRA-IASB mobile-DOAS setup follows the MAX-DOAS principle (Hönninger et al., 2004) and consists of a compact double Avantes spectrometer, recording scattered light spectra simultaneously in 2 channels, one at zenith and one at 30° off-axis. The spectral resolution is around 1.1 nm. The telescopes, yielding a 2.6° field of view, are assembled in an optical head and mounted on the roof of the car. They are connected to the spectrometers by 2 400 μm optical fibers. A GPS is used to geolocalise the measurements and the whole set-up uses the standard 12v car battery. The integration time is fixed at 5 ms and in order to increase the SNR, spectra series are averaged to a final spectrum, each 10 s. Thus, the spatial resolution of the measurements is around 138 m in case of a car speed of 50 km h$^{-1}$. Further details of the mobile-DOAS setup and the applied retrieval approach are discussed in Merlaud (2013a).

### 6.2 Mini MAX-DOAS observations

During the Brussels flight on 30 June, there was an overpass at 15:21 LT over the Uccle NDACC (Network for the Detection of Atmospheric Composition Change) candidate station (50.78° N, 4.35° E, 100 m ASL), where a mini-MAX-DOAS instrument is deployed (Ma et al., 2013; Gielen et al., 2014). The location of the station is plotted on Fig. 14. The operating instrument, a commercial system from Hoffmann Messtechnik GmbH which has been continuously running since 2011, is pointing towards the north (Brussels city center) with a field of view of 0.6° and has a spectral resolution of 0.6 nm FWHM. Each 15 minutes, the instrument performs a full MAX-DOAS scan, comprising 11 elevation angles, including a zenith measurement.

### 6.3 Correlative comparison of retrieved NO$_2$ VCDs

The APEX, mobile-DOAS, and mini-MAX-DOAS observations are targeting mainly NO$_2$ in the lower troposphere. To ensure comparability of the collocated data sets, the retrieval settings are harmonised as much as possible.

For the comparison with the mobile-DOAS data, a VCD is extracted from the generated APEX NO$_2$ distribution map for each mobile measurement. To reduce the noise and avoid sampling of the same APEX pixel due to traffic jams, each 2 subsequent retrievals are averaged. Both the mobile and APEX NO$_2$ VCD time series are plotted in Fig. 18.a and 18.b, for





Antwerp and Brussels respectively. Near-simultaneous observations with APEX and the mobile-DOAS system were pursued. In most cases, however, a $NO_2$ column at a certain geolocation is not sampled at exactly the same time and variability in local emissions and meteorology can lead to differences. In order to facilitate the interpretation of the $NO_2$ VCD time series, the time offset is plotted as well. A positive bias implies that a certain air mass was sampled earlier by the mobile-DOAS system than by APEX:

$$t_{offset} = t_{APEX} - t_{Mobile} \qquad (6)$$

For Antwerp and Brussels, the time series are in good agreement, both for low and high VCDs. In general, higher VCDs up to 3 x $10^{16}$ molec cm$^{-2}$ are measured in Antwerp, mainly originating from the industrial activities in the harbor area. The $NO_2$ VCDs measured in Antwerp by APEX and mobile-DOAS are respectively 1.8 and 1.6 x $10^{16}$ molec cm$^{-2}$ on average, and respectively 6.8 and 6.4 x $10^{15}$ molec cm$^{-2}$ for the Brussels data set. The mobile measurements are representative for the whole data set as the averages are close to the mean values for the full $NO_2$ VCD distribution maps, being 1.7 x $10^{16}$ and 7.7 x $10^{15}$ molec cm$^{-2}$ for Antwerp and Brussels, respectively. In general a positive bias of approximately 12 % and 6 % can be observed for the APEX retrievals, for Antwerp and Brussels, respectively. The larger bias for the Antwerp data set can be related to the larger time offset of up to 3 hours. The impact of variability in local emissions and meteorology is subsequently larger here. The parts without mobile observations in the time series are related to car stops or traffic jams. For the Brussels flight, additional efforts were done to minimise the time offset between the mobile and APEX observations. The commercial application flighradar24.com was used for this purpose, visualising the real-time position of the plane. The time offset is less than 1 hour and 2 overpasses can be identified which are synchronized both in space and time, one at 12:46 and one at 13:39 UTC. For both cases, the difference between the retrieved VCDs is 1.1 x $10^{15}$ molec cm$^{-2}$.

The overpass at 13:21 UTC over the Uccle mini-MAX-DOAS station, which is synchronised both in time and location, is also plotted on Fig. 18.b. As can be observed in Fig. 14, the station is located upwind of the city and other main sources, so an urban background was measured. Both measurements are in good agreement: A $NO_2$ VCD of 5.8 and 6.5 x $10^{15}$ molec cm$^{-2}$ was retrieved for the mini MAX-DOAS and APEX, respectively.

Scatter plots and linear regression analysis of the APEX and mobile-DOAS $NO_2$ VCDs (averaged by 2 subsequent retrievals) are given in Fig. 19.a and 19.b, for the Antwerp and Brussels data set, respectively. In total 521 observations for the Antwerp data set and 342 for Brussels were compared. The correlation coefficients are 0.84 and 0.85 for Antwerp and Brussels, respectively. Slopes are within 7% of unity and a larger intercept can be observed for the Antwerp data set. For Brussels, we see a slight positive bias for the mobile-DOAS in case of small VCDs, while it gets negative at higher VCDs. Efforts were done to ensure the comparability of the correlative data sets, but nevertheless the scatter can be largely explained by observation time differences in combination with variability of the $NO_2$ signal, sampling of different air masses due to the viewing geometry, differences in the sensitivity to $NO_2$, and instrumental and algorithmic conceptual differences and related errors and uncertainties.



## 7    Summary and conclusions

A retrieval scheme is presented to successfully infer the $NO_2$ VCD field over cities, based on the DOAS analysis of APEX level0-DC spectra. This is currently one of the few studies reporting on $NO_2$ horizontal distribution mapping at this scale and level of detail. APEX flights were performed above 3 of the largest urban areas in Belgium, being the cities of Antwerp and Liège (15 April 2015), and Brussels (30 June 2015). This study demonstrates that (1) the urban atmospheric $NO_2$ field can be mapped at high spatial resolution in a relatively short time frame, based on a systematic flightplan, (2) contributing local emission sources can be resolved, and (3) fine-scale structures within the detected plumes can be exposed. For example, Antwerp city, the harbor and the surrounding semi-rural area, covering approximately 350 km$^2$, have been mapped in less than 90 minutes. A spatial resolution of approximately 60 x 80 m$^2$ is obtained, after increasing the SNR to 2500 by a 20 x 20 spatial binning, with a superior geolocation accuracy of less than 3 m. The mapped $NO_2$ field shows that hotspots of enhanced $NO_2$, related to heavy traffic, industrial facilities, etc. can be identified and large emission sources can be distinguished.

The main sources in the Antwerp area (15 April 2015) appear to be related to (petro)chemical industry. The $NO_2$ levels range between 3 x $10^{15}$ molec cm$^{-2}$ in the west and 35 x $10^{15}$ molec cm$^{-2}$ in the east, downwind of the sources, with a VCD of 17.4 ± 3.7 x $10^{15}$ molec cm$^{-2}$ on average. The $NO_2$ levels observed in Liège (15 April 2015) range between 1 and 32 x $10^{15}$ molec cm$^{-2}$, with a mean of 13.3 ± 3.1 x $10^{15}$ molec cm$^{-2}$. The $NO_2$ levels in the Brussels area (30 June 2015) are on average 55% lower than in Antwerp with minima and maxima of 1 and 20 x $10^{15}$ molec cm$^{-2}$, respectively, and a mean VCD of 7.7 ± 2.1 x $10^{15}$ molec cm$^{-2}$. The $NO_x$ sources are mainly originating from traffic and are concentrated along the R0 ring road and junctions with the key highways. In order to quantitatively assess the APEX $NO_2$ retrievals, the Antwerp and Brussels data set are compared with correlative car mobile-DOAS measurements. Both data sets are in good agreement with correlation coefficients around 0.85 and slopes close to unity. However, APEX retrievals tend to be on average 12% and 6% higher for Antwerp and Brussels, respectively.

The APEX instrument was initially not designed for trace gas retrieval applications. Despite its outstanding spatial resolution and georeferencing when compared to other imaging systems, and the benefit of being radiometrically calibrated periodically (at least once a year), this study has revealed some limitations related to the spectral performance, i.e. spectral resolution, sampling rate and robustness of the slit function in operational conditions. The economised spectral information and discussed instabilities lead to (1) additional mandatory steps in the retrieval approach, e.g. accurate wavelength calibration and the selection of different reference spectra, (2) a higher detection limit, and (3) higher uncertainties in the retrieval approach. The error budget assessment indicates that the overall error $\sigma_{VCD}$ on the retrieved $NO_2$ VCDs is on average 21%, 23% and 28% for the Antwerp, Liège and Brussels data set, respectively. Low VCD retrievals are mainly limited by noise (error on the DOAS fit or 1-sigma slant error), while large retrievals are mainly limited by both the slant error as systematic errors, e.g. related to the AMF computations.





High resolution quantitative information about the atmospheric $NO_2$ horizontal variability is currently rare, but can be very valuable for (air quality) studies in urbanised areas. Airborne observations of the $NO_2$ field complement and bridge the gap between local point observations of ground stations, global monitoring by spaceborne instruments, and model output. Recently, new APEX flights were performed over Antwerp, Belgium as well as over Berlin, Germany. Revisiting Antwerp

in the framework of the BUMBA project (Belgian Urban $NO_2$ Monitoring Based on APEX remote sensing) will (1) further improve the characterisation of the (temporal) $NO_2$ distribution in the city and (2) further consolidate the developed retrieval algorithm. The APEX flights over Berlin were performed in the framework of the ESA-EUFAR supported AROMAPEX campaign, clustered in the AROMAT project (Airborne ROmanian Measurements of Aerosols and Trace gases). Two additional imagers, AirMAP (Airborne imaging DOAS instrument for Measurements of Atmospheric Pollution; Schönhardt

et al., 2015; Meier et al., 2016) and SWING (Small Whiskbroom Imager for atmospheric composition monitoriNG; Merlaud et al., 2013b), were simultaneously operated from a Cessna from the Free University of Berlin, providing a unique data set for intercomparison purposes. These new data sets are currently under analysis, taking into account the findings and lessons learned from the research described in this paper.

*Acknowledgements.* The Belgian Federal Science Policy Office is gratefully appreciated for funding the presented work and

15 the BUMBA (Belgian Urban $NO_2$ Monitoring Based on APEX remote sensing) project, through the STEREO III Research Programme for Earth Observation (SR/00/310). The authors also wish to express their gratitude to the other partners of the BUMBA project, i.e. the Flemish Institute for Technological Research (VITO) and the Belgian Interregional Environment Agency (IRCEL-CELINE). The authors wish to thank Gerrit Kuhlman and Dominik Brunner from the Swiss Federal Laboratories for Materials Science and Technology (EMPA) and Steffen Dörner of the Max-Planck-Institute for Chemistry

(MPIC) for fruitful discussions.

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



**Table 1.** APEX spatial performance and instrumental specifications.

| Spatial performance (at 6100 m AGL) | |
|---|---|
| **Spatial CCD** | 1000 detectors |
| **FOV (across-track)** | 28° |
| **Swath width** | 3000 m |
| **IFOV (across-track)** | 0.028° |
| **Spatial resolution (across-track)** | 3 m |
| **Spatial resolution (along-track)** | 4 m |
| **Other** | |
| **Plane speed** | 72 m.s$^{-1}$ |
| **Integration time** | 58 ms |
| **APEX total mass** | 354 kg |

**Table 2.** Acquired data sets and flight characteristics. Wind and temperature data are collected from weather stations of the Royal Meteorological Institute (RMI) and averaged over the time of flight. Population data from 1 January 2015, retrieved from Statistics Belgium: http://statbel.fgov.be/, last access: 15 January 2016. For each flight, the day number of the year 2015 is mentioned between brackets and will be used further in the manuscript to refer to the different flights.

| | Brussels | Antwerp | Liège |
|---|---|---|---|
| **Date** | 30-06-2015 (181) | 15-04-2015 (105) | 15-04-2015 (105) |
| **Flight time LT (UTC + 2)** | 14:43 - 16:04 | 10:06 - 11:30 | 11:55 - 12:18 |
| **# flightlines** | 8 | 9 | 3 |
| **Flight pattern (Heading °)** | 0 - 180 | 0 - 180 | 40 - 220 |
| **SZA (°)** | 29.7 - 38.6 | 60.4 - 49.6 | 46.0 - 44.1 |
| **Wind direction (°)** | 125 | 235 | 240 |
| **Wind speed (Bft)** | 2 | 3 | 3 |
| **Temperature (°C)** | 27.2 | 18.7 | 20.8 |
| **PBL height (m)** | 1200 | 500 | 700 |
| **Lat (°N) / Long (°E)** | 50.8 / 4.4 | 51.2 / 4.4 | 50.6 / 5.6 |
| **Terrain altitude (m ASL)** | 76 | 10 | 66 |
| **Total population** | 1.175.173 | 513.570 | 195.968 |
| **Population density (#/km$^2$)** | 6751 | 2496 | 2828 |





**Table 3.** APEX spectral performance for the $NO_2$ calibration window. Both nominal (laboratory performance) and effective (in-flight performance) parameters are provided. Due to the wavelength- and pixel location-dependency, FWHM and shift are provided for 490 nm, i.e. the middle of the analysis window, and for the nadir looking detector element of the pushbroom sensor.

| Spectral performance for $NO_2$ calibration window | |
|---|---|
| **Spectral interval** | 370 - 600 nm |
| **Spectral detectors** | 249 (unbinned mode) |
| **Nominal FWHM** | 1.5 nm |
| **In-flight FWHM** | > 2.8 and < 3.3 nm |
| **Nominal spectral shift from CW** | < 0.2 nm |
| **In-flight spectral shift from CW** | > 0.05 and < 0.8 nm |
| **Spectral sampling interval (SSI)** | 0.9 nm |
| **Sampling rate** | 3.1 to 3.6 pixels per FWHM |

**Table 4.** Main DOAS spectral fitting analysis parameters for $NO_2$ slant column retrieval.

| Parameter | Settings |
|---|---|
| **Wavelength calibration method** | Reference solar atlas (Chance and Kurucz, 2010) - Gaussian |
| **Calibration interval** | 370 – 600 nm (5 sub-windows) |
| **Fitting interval** | 470 – 510 nm |
| **Cross-sections** | |
| $NO_2$ | Vandaele et al. (1998),  298 K |
| $O_4$ | Hermans et al. (2003) |
| **Ring effect correction method** | Chance and Spurr (1997) |
| **Resol** | Small diff. in spectral resolution |
| **Polynomial term** | Order  5 |
| **Intensity offset correction** | Slope |



**Table 5.** Sensitivity study with varying input parameters in the radiative transfer model based on 73000 observations of the Antwerp data set. For the albedo, sun and viewing geometry, each time 2 scenarios are provided based on the μ +/- 1σ level and corresponding TAMFs are derived by BAMF integration along a mean AURORA $NO_2$ profile. For the a priori $NO_2$ profile shape sensitivity study, 2 scenarios are given first: integration along (1) local $NO_2$ vertical profiles $A_{interp}$, interpolated on the AURORA model grid and (2) a fixed AURORA $NO_2$ profile $A_{harbor}$ over a polluted area. TAMFs for the Antwerp data set are calculated based on these 2 scenarios, as well as the variability. Secondly, TAMFs are computed by BAMF integration along (1) local $NO_2$ vertical profiles $A_{interp}$, interpolated on the AURORA model grid and (2) a well-mixed $NO_2$ box profile of 0.5 / 1 km height in the lowest layer.

| RTM parameter | Parameter μ - 1σ | Parameter μ + 1σ | TAMF μ - 1σ | TAMF μ + 1σ | TAMF variability |
|---|---|---|---|---|---|
| **Albedo** | 2% | 8% | 1.3 | 2.2 | 65% |
| **RAA** | 37.1° | 151.1° | 1.9 | 1.9 | 4% |
| **VZA** | 4.4° | 11.5° | 1.9 | 1.9 | 1% |
| **SZA** | 51.2° | 58° | 1.9 | 2.0 | 6% |
| **$NO_2$ profile** | $A_{interp}$ | $A_{harbor}$ | 1.9 | 1.8 | 8% |
| **$NO_2$ profile** | $A_{interp}$ | $Box_{0.5km}$ | 1.9 | 1.8 | 7% |
| **$NO_2$ profile** | $A_{interp}$ | $Box_{1km}$ | 1.9 | 1.9 | 1% |

**Table 6.** Error budget analysis based on 73000 retrieved $NO_2$ VCDs of the Antwerp data set. Typical relative errors (percent) and absolute errors (x $10^{15}$ molec $cm^{-2}$ for $\sigma_{DSCD}$, $\sigma_{SCDref}$ and $\sigma_{VCD}$) are provided for small (< 33[th] percentile or < 1.4 x $10^{16}$ molec $cm^{-2}$), medium (33[th] to 66[th] percentile or 1.4 to 2.0 x $10^{16}$ molec $cm^{-2}$) and high (> 66[th] percentile or > 2.0 x $10^{16}$ molec $cm^{-2}$) $NO_2$ VCD retrievals, respectively. The last column gives the errors for all retrieved VCDs together.

| Error source | small VCD | Medium VCD | High VCD | Total VCD |
|---|---|---|---|---|
| $\sigma_{DSCD}$ | 40% (3.8) | 23% (3.9) | 15% (3.9) | 22% (3.9) |
| $\sigma_{SCDref}$ | 19% (1.8) | 11% (1.8) | 7% (1.8) | 10% (1.8) |
| $\sigma_{TAMF}$ | 15% (0.3) | 15% (0.3) | 15% (0.3) | 15% (0.3) |
| $\sigma_{VCD}$ | 29% (2.8) | 21% (3.5) | 18% (4.7) | 21% (3.7) |





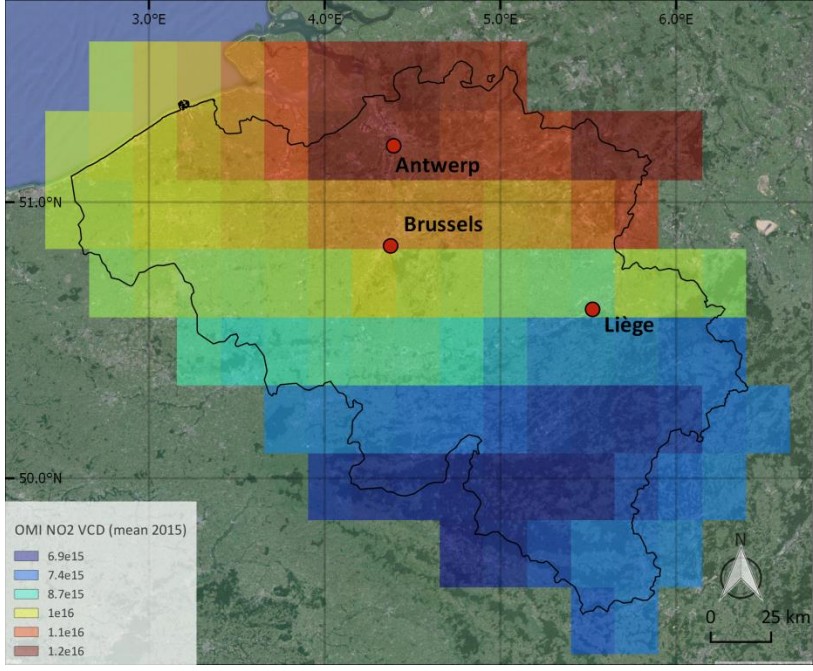

**Figure 1.** OMI annual NO$_2$ VCD map for Belgium, 2015 and overview of the 3 Belgian cities where APEX flights were performed (Google, TerraMetrics, Giovanni, NASA GES DISC, OMNO2d, units in molec cm$^{-2}$).



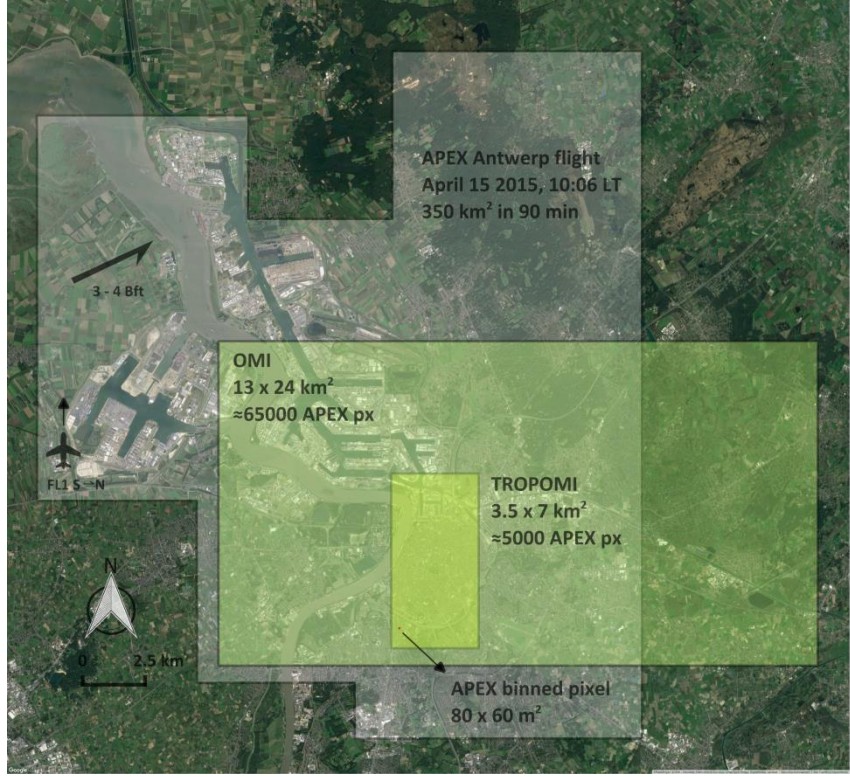

**Figure 2.** Details of the APEX flightplan over Antwerp on 15 April 2015 and comparison with a spaceborne nadir OMI and TROPOMI pixel (Google, DigitalGlobe).



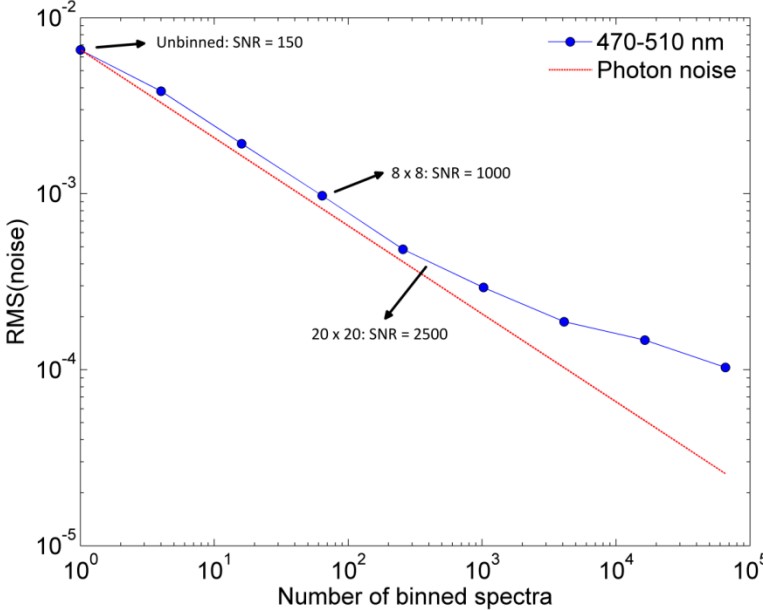

**Figure 3.** Allan plot illustrating the impact of spatial binning of the raw spectra on the RMS of the noise, plotted on a double logarithmic scale.




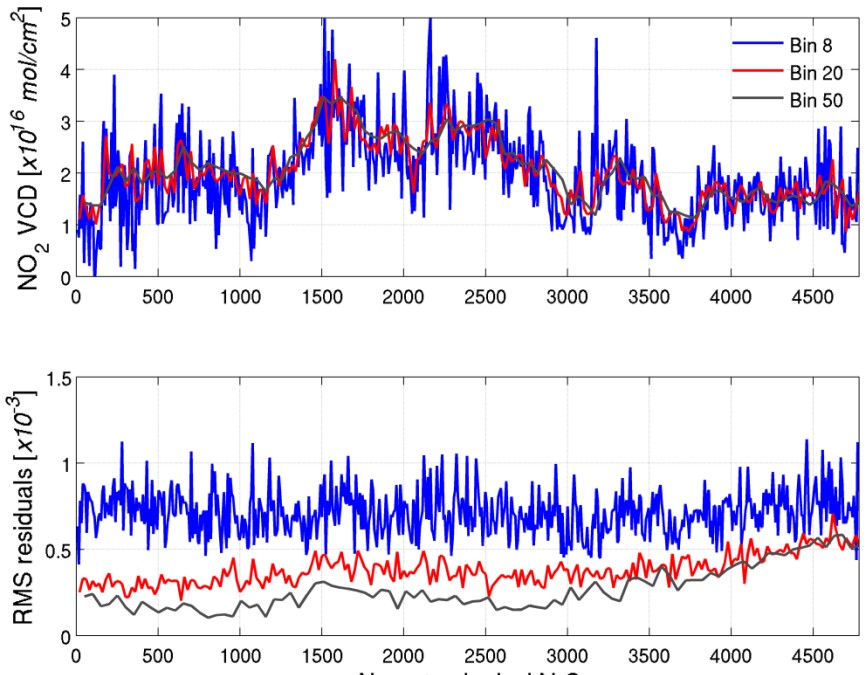

**Figure 4.** Time series of **a)** NO$_2$ VCD and **b)** RMS error of DOAS fits for 3 different levels of aggregated spectra, i.e. 8 by 8, 20 by 20, and 50 by 50 pixels. The VCDs are retrieved from an overlapping column on flightline 8 of the Antwerp data set and are plotted from north to south.





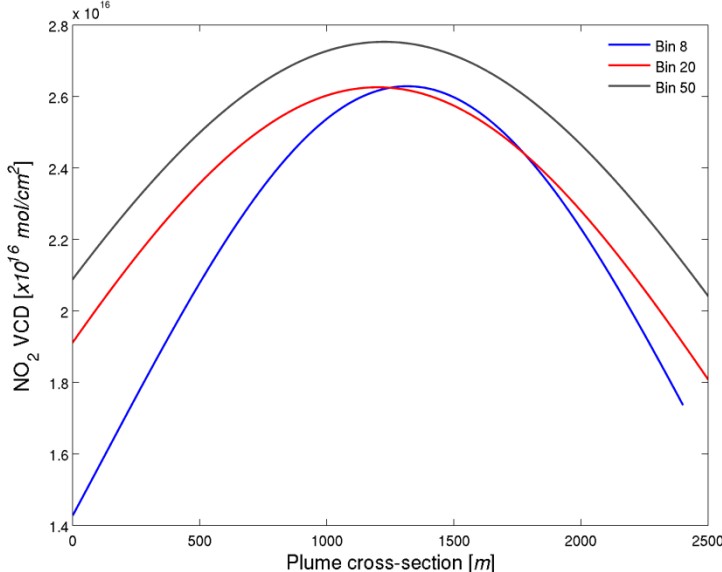

**Figure 5.** Gaussian fit on the $NO_2$ VCDs from an orthogonal horizontal profile taken through a $NO_2$ plume, for 3 different binning levels, i.e. $8^2$, $20^2$ and $50^2$ pixels.





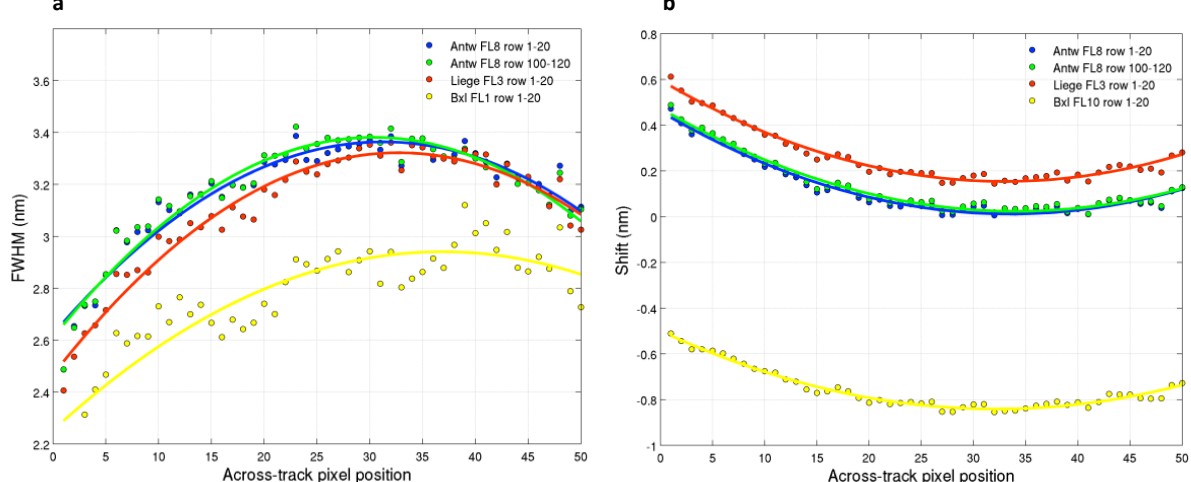

**Figure 6.** Dependency of **a)** the spectral resolution (FWHM) and **b)** the spectral shift (smile) on the across-track scanline pixel position and time of acquisition. The spectral calibration is performed at 490 nm, i.e. the middle of the analysis window.





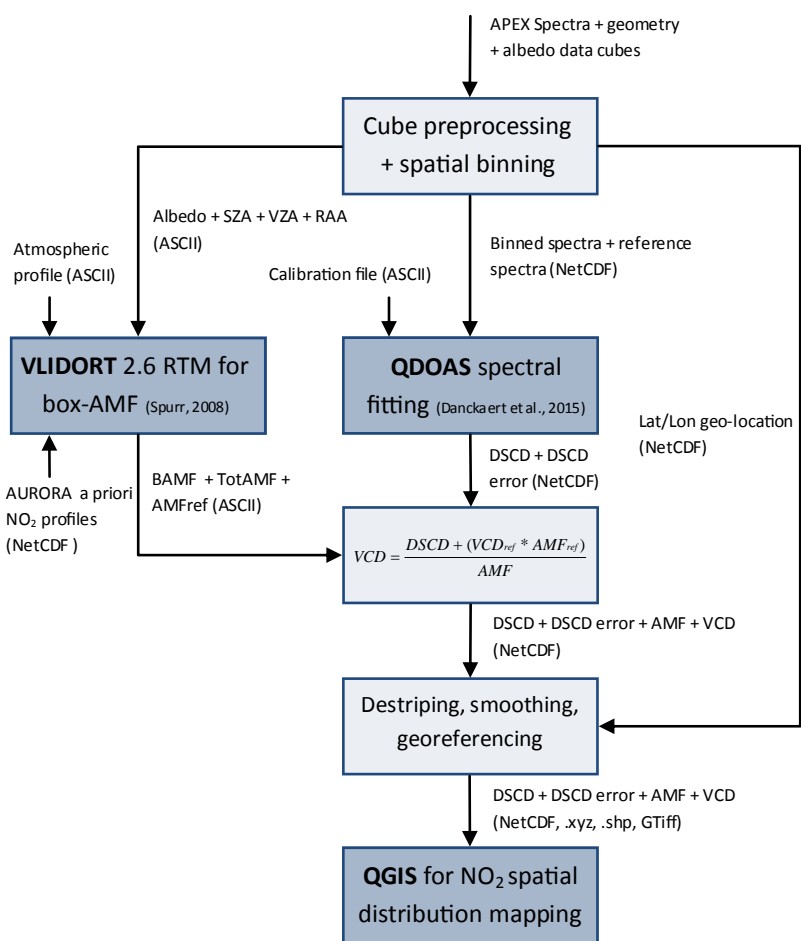

**Figure 7.** Flowchart of the APEX NO$_2$ VCD retrieval algorithm.





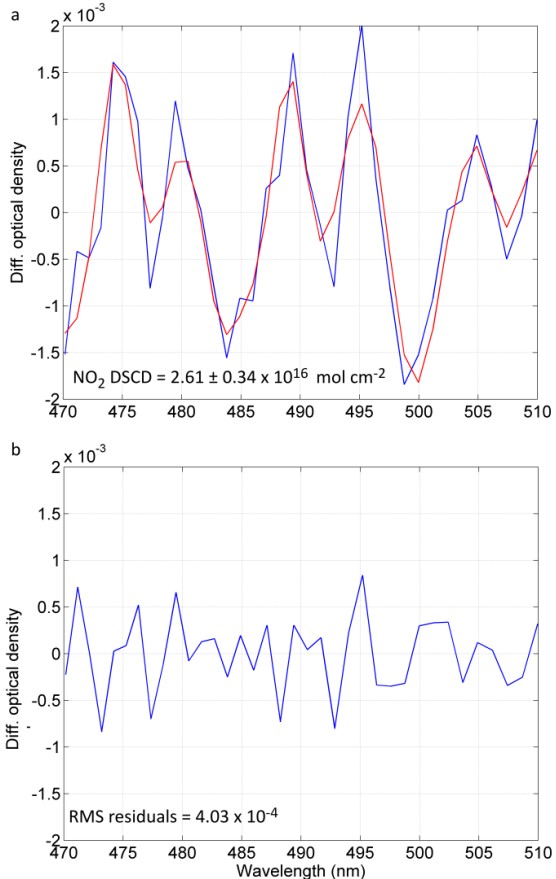

**Figure 8.** Typical DOAS fit with **a)** red line, corresponding to the NO$_2$ molecular cross-section, convolved with the instrument slit function and scaled to the detected absorption in the measured spectrum (blue line) and **b)** the remaining residuals of the spectral fit.





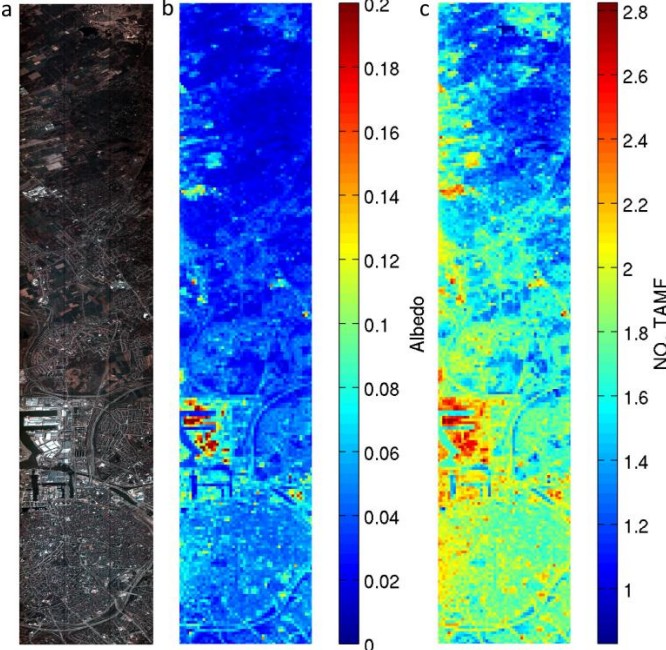

**Figure 9. a)** APEX true color composite, **b)** APEX albedo level 2 product and **c)** computed TAMFs, for flightline 8 of the Antwerp data set (15 April 2015). The strong dependency of the AMF to the albedo can be observed.





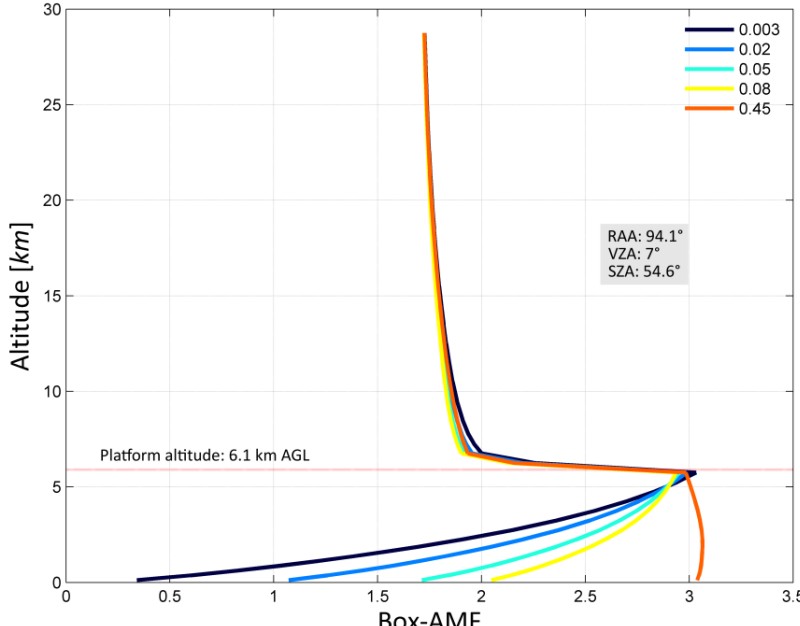

**Figure 10.** BAMF profiles illustrating the vertical sensitivity of the APEX instrument to NO$_2$. The high impact of the surface albedo, mainly on the lowest atmospheric layers, is shown based on 5 different scenarios, ranging from low to high albedo. Scenarios are based on the minimum, μ - 1σ, mean, μ + 1σ and maximum albedo in the Antwerp data set.





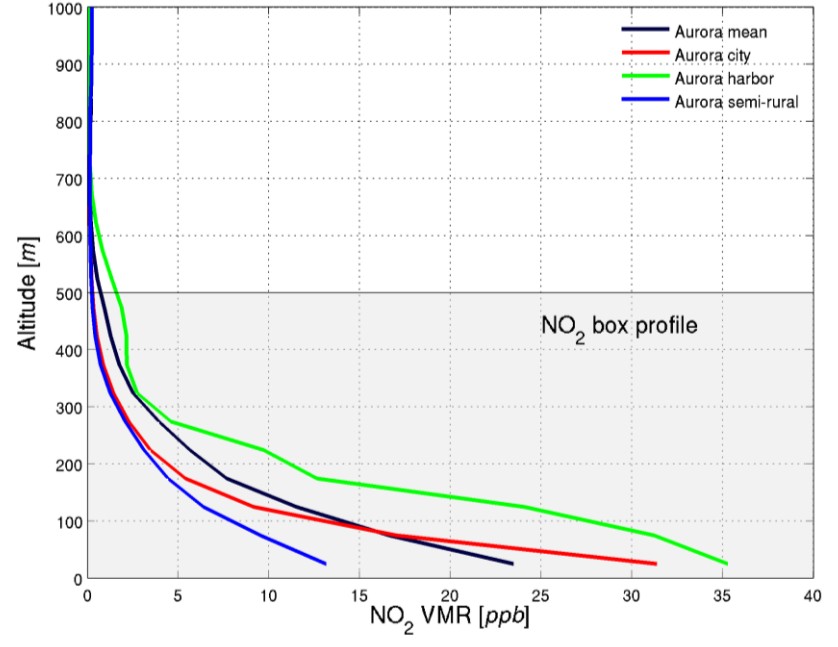

**Figure 11.** Representative AURORA a priori $NO_2$ profiles, used for the Antwerp data set RTM calculations. A simple $NO_2$ box profile of 500 m height which is well-mixed in the boundary layer is used as well in the sensitivity study.





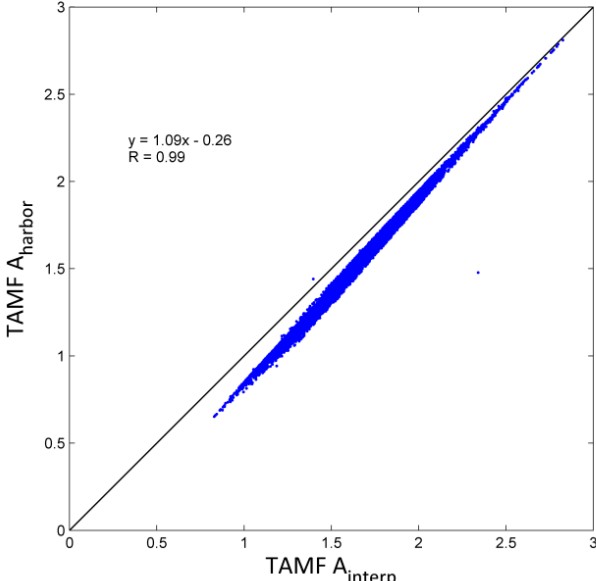

**Figure 12.** Scatter plot and linear regression analysis for the TAMF computation, based on 2 scenarios: integration of the BAMFs along (1) local $NO_2$ vertical profiles $A_{interp}$, interpolated on the AURORA model grid and (2) a fixed AURORA $NO_2$ profile $A_{harbor}$, with high $NO_2$ concentrations related to industrial sources.





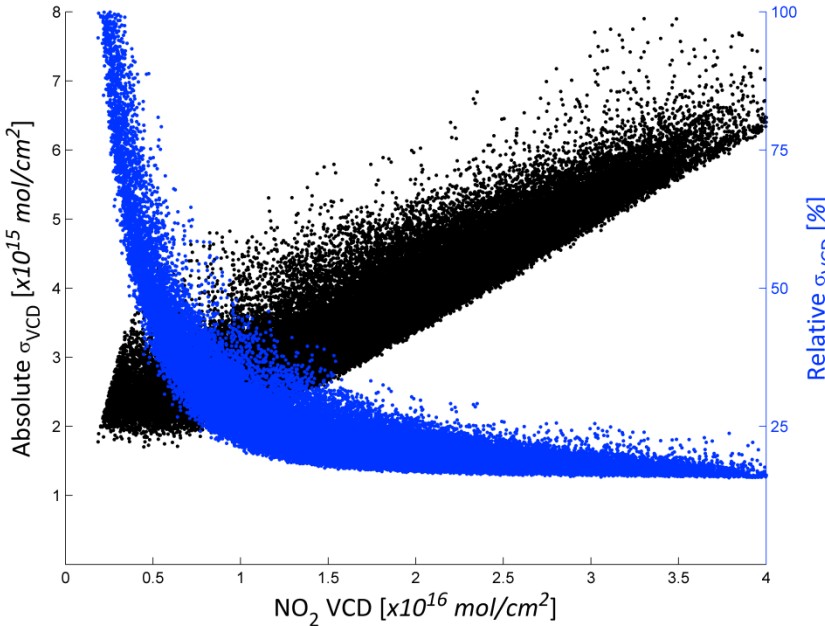

**Figure 13.** Overall absolute (black dots) and relative errors (blue dots), $\sigma_{VCD}$, on the retrieved $NO_2$ VCDs, based on the Antwerp data set.





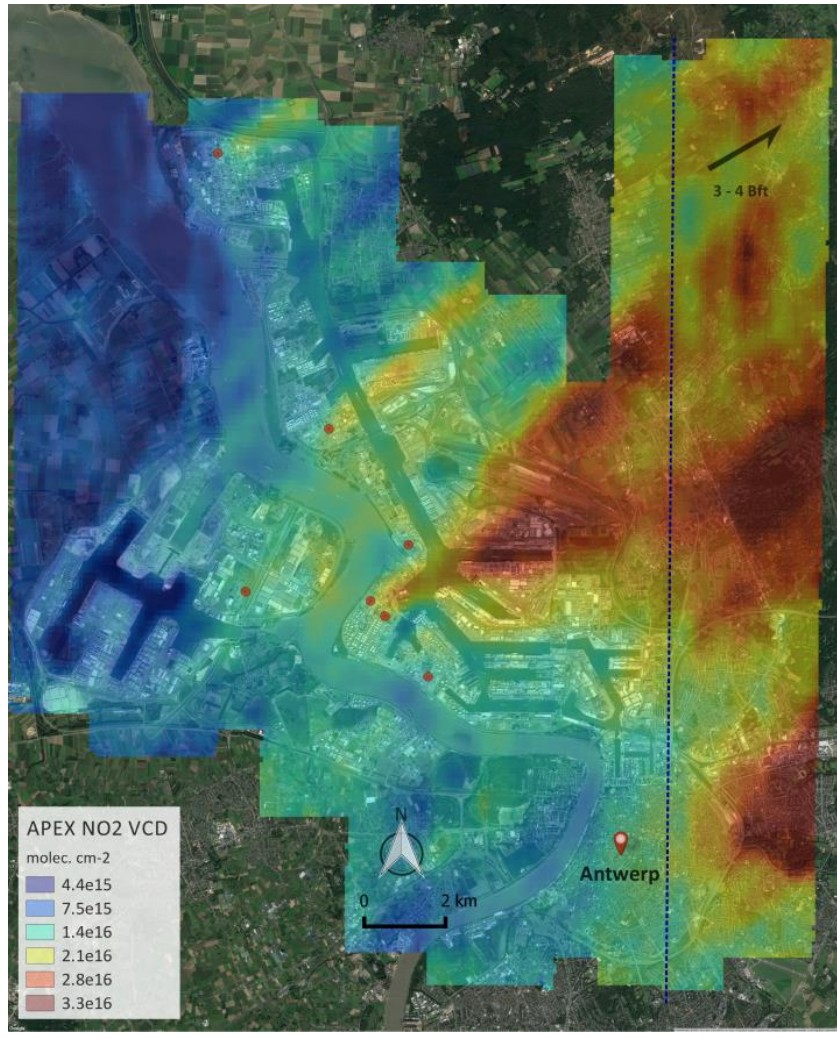

**Figure 14.** Retrieved NO$_2$ VCD field for Antwerp (15 April 2015) (Google, TerraMetrics). Red dots indicate the chimney stacks, emitting more than 25 kg of NOx per hour, according to the emission inventory of the Belgian Interregional Environment Agency. The blue vertical line indicates an along-track profile, for which the NO$_2$ DSCD and VCD time series are plotted in Fig. 15.





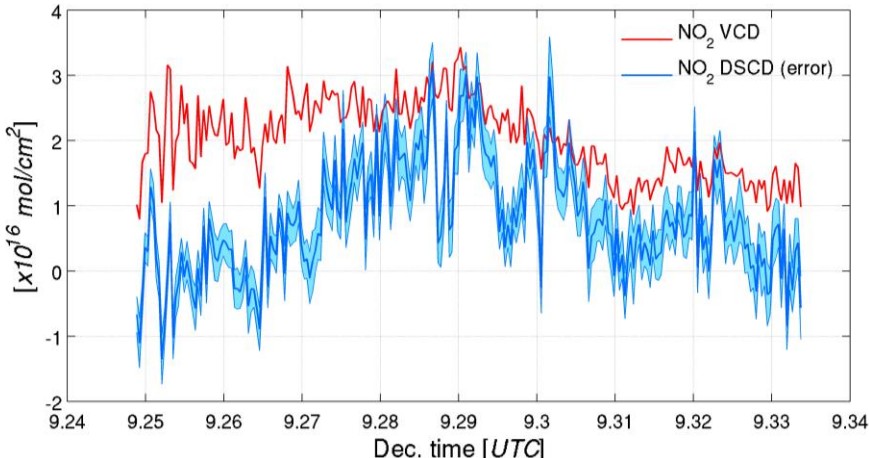

**Figure 15.** NO$_2$ DSCD and VCD time series for an along-track profile from north to south taken on flightline 8 of the Antwerp data set (15 April 2015). The retrieved NO$_2$ signal is well above the detection limit. The profile, crossing the main plume from the harbor and the city center, is indicated by a dashed (blue) vertical line in Fig. 14.





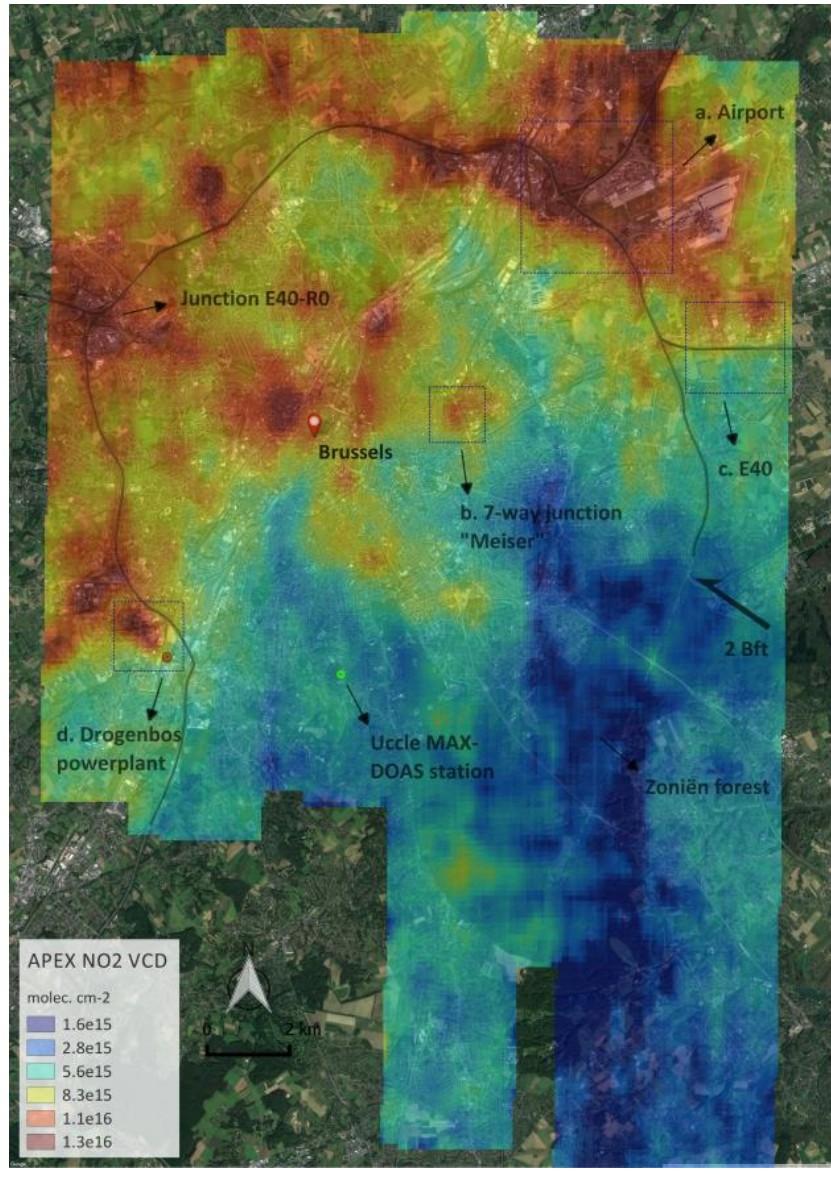

**Figure 16.** Retrieved NO$_2$ VCD field for Brussels (30 June 2015) (Google, TerraMetrics). Blue squares indicate 4 NO$_2$ VCD hotspots, highlighted in Fig. 17. The green dot is the location of the Uccle MAX-DOAS station.





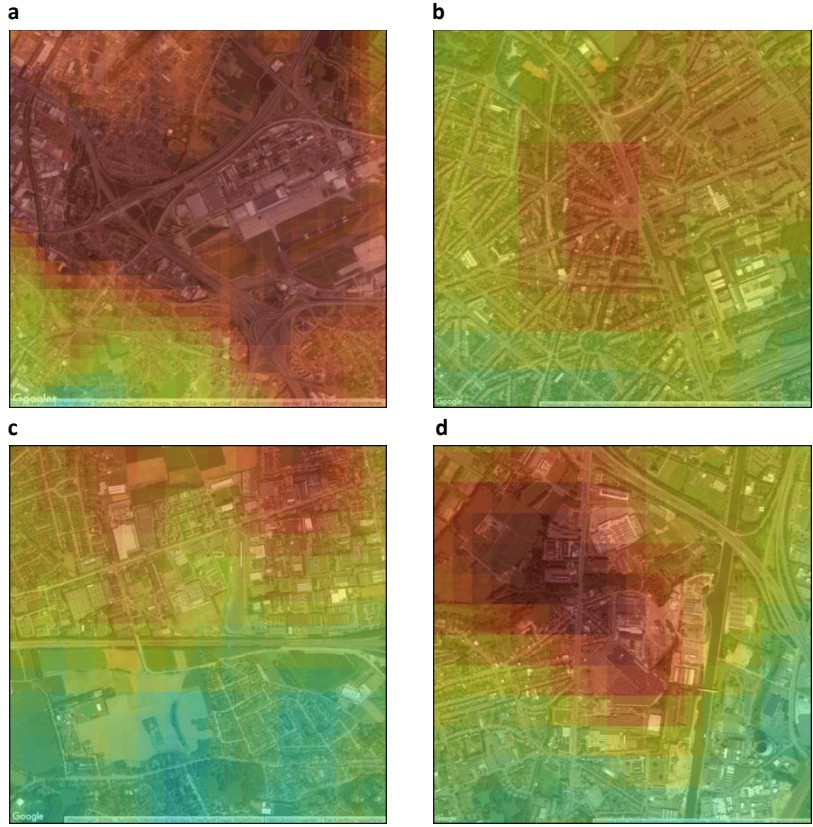

**Figure 17.** Enlargement of 4 peculiar $NO_2$ VCD hotspots in the Brussels data set (30 June 2015): **a)** major junction between the ring road R0 and the E19, and Brussels international airport; **b)** junction "place Meiser", close to the city center; **c)** eastern part of the E40 highway; **d)** gas turbine Drogenbos powerplant. The locations of the 4 zooms are indicated in Fig. 14 by blue squares (Google, TerraMetrics).




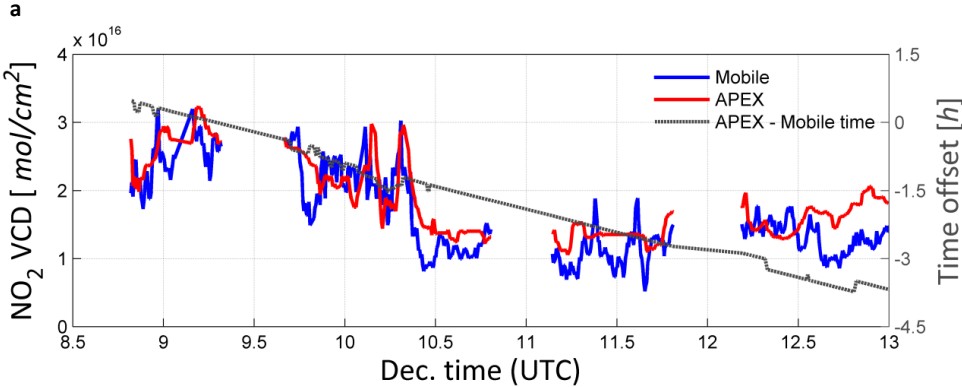

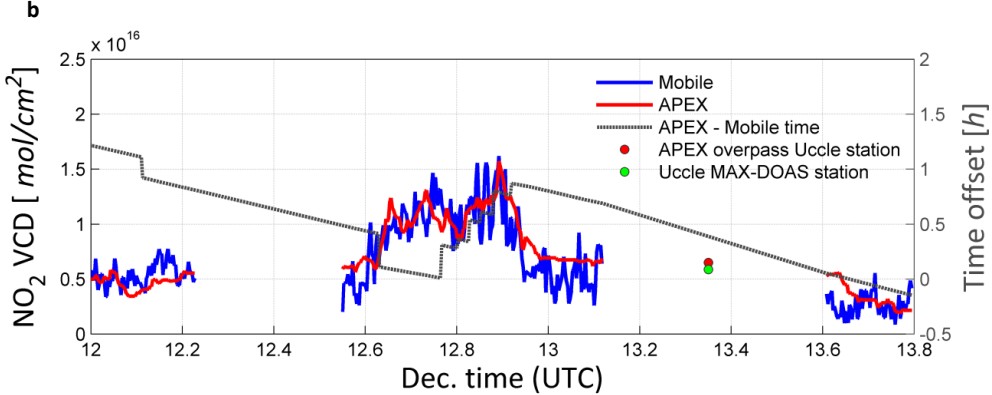

**Figure 18.** APEX and Mobile-DOAS $NO_2$ VCD time series for **a)** the Antwerp flight (day 105) and **b)** the Brussels flight (day 181), respectively. The time offset between APEX and mobile-DOAS observations is plotted in dark grey. The $NO_2$ VCDs, measured by APEX (red dot) and the mini-MAX-DOAS (green dot) at the overpass over the Uccle station (13:21 UTC) are indicated on Fig. 18.b.





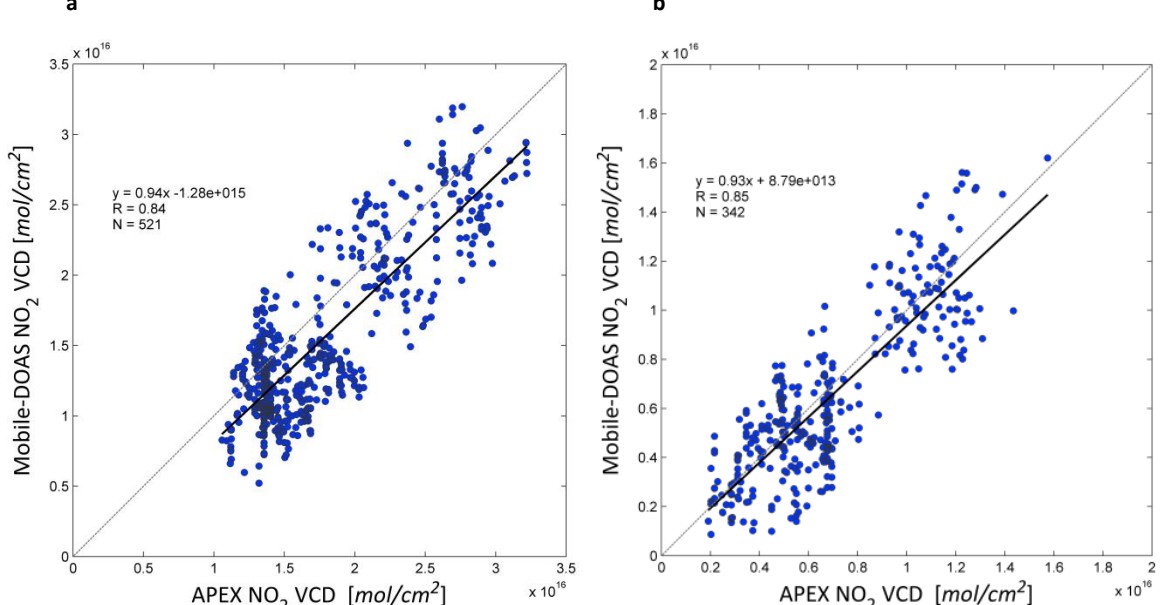

**Figure 19.** Scatter plot and linear regression analysis of the position-synchronised NO$_2$ VCDs, retrieved from APEX and mobile-DOAS for **a)** the Antwerp flight (day 105) and **b)** the Brussels flight (day 181), respectively.