# Peer review of "High resolution mapping of the NO2 spatial distribution over Belgian urban areas based on airborne APEX remote sensing"

_Atmospheric Measurement Techniques, 2016_

## Referee Comment (RC1) · G. Kuhlmann (Referee) · 19 Jan 2017

**General comments**

The paper describes the retrieval of high resolution NO2 maps from the airborne imaging spectrometer APEX over three cities in Belgium (Antwerp, Brussels and Liege). The authors develop a new retrieval algorithm for APEX similar to the algorithm described by Popp et al. (2012) but with several improvements such as better characterization of the influence of spatial binning and reference NO2 VCD on the retrieval as well as a correction term for in-flight changes of spectral resolution ("resolution cross section"). They also estimate VCD errors, which were not included in the paper by Popp et al.,

and compared their measurements with car mobile-DOAS measurements. The authors conclude that an airborne imager can be used for studying the spatial distribution of NO2 in urban areas.

The authors present new ideas, tools and data and their conclusions are mostly substantial. The topic is well within the scope of AMT. The scientific methods are valid and mostly clearly described. Some revisions are required, in particular, in Sections 3.1 and 3.2 which contain some small inaccuracies and/or not fully understandable results (see special comments). The paper is generally well written and the trail of ideas is understandable, but additional copy-editing is necessary.

**Special comments**

Section 1 (Introduction)

The authors state that one objective is to access APEX's capabilities for mapping NO2 in a "cost-effective way" (p. 2, l. 13ff). However, the cost effectiveness of APEX is not discussed in the paper. Thus, I suggest revising the objective or adding a section/paragraph on the cost effectiveness of APEX.

Section 3.1

The authors describe the data as "time series" (p. 7, l. 4f and Fig. 4), but since the spatial dimension should be more important than temporal dimensions, I think a better term would be "along-track profile" as used in Fig. 14.

The description of the smoothing effect and conclusion from this analysis are difficult to understand (p. 7, l. 6ff and Fig. 5). The authors should consider showing the raw

data used for the fits in Fig. 5 and mark the location of the x-axis of Fig. 5 in Fig. 14.

Section 3.2

Paragraphs 3 and 4 need to be revised, because the conclusions are based on a wrong premise. APEX has no absolute pressure stabilization but is kept at 200 hPa above ambient pressure (Kuhlmann et al. 2016). Thus, CW positions depend on the ambient pressure at flight altitude during data acquisition and spectral shifts can occur between different campaigns. I think this is the most likely explanation for the differences found by the authors between the three datasets such as the shift between Antwerp and Brussels which were not flown on the same day.

Furthermore, the CW accuracy of 0.2 nm does not include the spectral smile, which is defined as the range of wavelength shifts in across-track direction. Schaepman et al. (2015) state the spectral smile is smaller than 0.35 px (px = nm smile per nm nominal FWHM) which is about 0.5 nm at 490 nm. This is in good agreement with the spectral smile (about 0.4 nm and not 0.8 nm) the authors show in Figure 6b.

APEX's in-flight spectral calibration of is described by D'Odorico et al. (2011) and Kuhlmann et al. (2016). The authors should consider citing these papers:

D'Odorico et al.: Performance assessment of onboard and scene-based methods for Airborne Prism Experiment spectral characterization, doi: https://doi.org/10.1364/AO.50.004755, 2011.

Kuhlmann et al.: An Algorithm for In-Flight Spectral Calibration of Imaging Spectrometers, doi: https://doi.org/10.3390/rs8121017, 2016.

Section 4

Since a "synthetic resolution cross section" (p. 9, l. 23) is not commonly used in DOAS fits, the authors should provide additional explanations or a reference.

While the differential approach reduces the impact of systematic instrumental stabilities and Fraunhofer lines, it does not necessarily neutralize them (p. 10, l. 10).

The authors state that spatial resolution of the NO2 map is 60x80m2. However, for visualization the maps are spatially smoothed by a Savitzky-Golay filter (p. 14, l. 9f). The authors should add sentence explaining how much this additional smoothing reduces the effective resolution of the NO2 map.

The DOAS fitting error is not a random error, because it also contains systematic errors, e.g. from the parametrization (p. 15, l. 5).

The authors should add the magnitude of the reference VCD used for the three campaigns.

Section 5/6

I think the discussion of the results need additional data for supporting the conclusions. The authors should consider providing a time series of NO2 concentrations for the three sites. These data would help to interpret the bias expected between adjacent flight lines (p. 17, l. 1f) and the difference between the cities (p. 18, l. 5ff). In particular, the lower NO2 levels in Brussels might be also explainable by synoptic differences and the diurnal NO2 cycle (Antwerp was flown 11 LT and Brussels at 15 LT). In addition, additional labels on maps would help to follow the explanations (Figs. 14, 16 and 17). An NO2 time series would also help interpreting the bias between APEX and mobile-DOAS measurements (p. 20, l. 12ff).

The authors might consider adding the route of the mobile measurements in Figures 14 and 16. The results for Liege are not shown and only briefly discussed in the paper with the reasoning that the results are similar to the other findings (p. 17, l. 19ff). However, since the Liege dataset is mentioned throughout the paper, I think the NO2 map for Liege should be added to the paper or the supplement for completeness.

Section 7

The authors state that APEX can expose fine-scale structures in plumes (p. 21, l. 7). I think this should be specified to "city plume" or "urban plume", because fine-scale structures in stack plumes are not visible in the maps. The authors should add the information compared to what the geolocation accuracy is superior (p. 21, l. 10).

Table 5

The authors should consider splitting Table 5 in two tables, because the headings for the last three rows (NO2 profile) seem not to match with the table content. In result, the quite long caption would be shorter, as well.

Table 6

I think the last column is probably showing the mean or median error for all VCDs and not the error of the total VCD.

**Technical corrections**

p. 4, l. 18: "bulk" -> "majority"?

p. 7, l. 1f: What is the column co-located with?

p. 13, l. 3: "compensation factor" -> "enhancement factor" or "air mass factor"

p. 17, l. 20: 11u30 -> 11:30

p. 20, l. 24: It is not clear what is meant by: "averaged by 2 subsequent retrievals".

p. 20, l. 29: "Gerrit Kuhlman" -> "Gerrit Kuhlmann"

Table 1: I would suggest using "pixels" instead of "detectors" in the second row.

Table 2: "Date" -> "Date (day of year)" and "0-180" -> "0, 180" in row flight pattern

Table 4: Is "slope" a polynomial of degree 1?

Figure 6: "performed at" -> "shown for"?

Figure 15/18: X-axis label: "Dec. time" -> "decimal time"?

---

## Referee Comment (RC2) · Anonymous Referee #2 · 10 Feb 2017

Comment on *High resolution mapping of the NO2 spatial distribution over Belgian urban areas based on airborne APEX remote sensing* by Frederik Tack et al.

**General Comments**

The paper by Tack et al., describes mapping of NO2 over Belgian cities using the APEX airborne spectral imager, an instrument that was not initially designed for the retrieval of trace gases. The authors present and their retrieval algorithm and discuss instrument performance during measurement flights. Airborne NO2 results are compared with collocated car observations and yield good agreement.

Overall the paper is well written and arguments are mostly easy to follow. The manuscript presents a good summary of the data analysis and NO2 results fulfills the in the introduction stated objective to assess the technical and operational capabilities of APEX NO2 mapping. However, a more detailed analysis of the instrument in-flight performance, as well as a discussion on possible improvements is missing, which would be very fitting for the scope of AMT (see also specific comments).

**Specific Comments**

Abstract: Please state right away that Liege results are not further discussed or leave out.

Section 1: Typically airborne experiments are cost intensive field measurements compared to e.g. car measurements. Please explain why APEX measurements are described as "cost effective".

Section 2.1: p.4, line 21: Please define typical altitude, speed, integration time.

Section 2.2: p.5, line 28: should say full NO2 columns below the plane.

Section 3.1: p.6, line 14: How is the 58ms integration time chosen? Are there problems with saturated scans at times?

Section 3.1: p.6, line 21: It is not clear which setting are used for the noise analysis, since Sect. 3.2 talks about calibration and analysis.

Section 3.2: p.8: The strong variability of the instrument slit function is indeed a challenge for the DOAS analysis. Here further details on the instrument would be interesting, e.g., are there time traces of the instrument's pressure and temperature? Could these be correlated with the behavior of the slit function? Or even used to improve data analysis? The authors have optimized the DOAS analysis given the current state of the spectra, but recommendations of technical instrument improvements (if possible) or looking into exploiting technical in-situ data (if available) are missing, which would be well suited for the AMT audience.

Section 4.3.1: Is there in-situ temperature and pressure data available or ground measurements that could be interpolated? Please discuss choice of US Standard Atmosphere.

Section 4.3.2: p.12, line 29: "similar statistics" here and elsewhere could go into an Appendix.

Section 4.3.2: p.13 and Section 4.6, p.15, line 27: Based on Fig. 16, the station in Uccle is in a semi polluted area, so any AOD measured there will not be representative for heavily polluted areas. The error on the aerosol effect seems underestimated. Please include an assessment using higher AODs.

Section 5, p.18, line 4: dSCDs around 0 are not "well above detection limit". Please be more specific with the respective statement (also in Fig. 15).

All Tables: Tables typically have a short caption. Please move explanations to text.

Table 1: Please choose detectors or pixel consistently. What does the plane speed relate to? The section "other" seems very random. I suggest integration in text where fitting.

Table 3: Can you comment on why the in-flight FWHM is significantly larger than nominal?

Table 4: Please explain "resol"

Fig. 3: Is the reference noise taken into account here?

Fig. 6: include description of lines in legend or caption.

**Technical corrections**

Please review manuscript for rules on spelling out digits.

p.3, line 1: Figs. 1 and 2

p.9, line 1 (SCDi) is already introduced

p.9 Eqs. 2 and 3: "ref" should be subscript

p.9, line 19: exchange word order of "wavelength" and "lower"

p.11ff: Please be consistent with either Box-AMF or box-AMF.

p.15, end of line 22: "the" is missing

p.17, line 20: 11:30 UTC

Fig. 4: labels a) and b) are missing

Fig. 6: Check rules on including units: "()" or "[]", cursive or not?

---

## Author Comment (AC1) · 3 Apr 2017

The comment was uploaded in the form of a supplement:
http://www.atmos-meas-tech-discuss.net/amt-2016-400/amt-2016-400-AC1-supplement.pdf

---

## Author Response (AR1)

04/04/2017

Dear editor,

With regard to the manuscript:

Journal: AMT
Title: High resolution mapping of the $NO_2$ spatial distribution over Belgian urban areas based on airborne APEX remote sensing
Author(s): F. Tack et al.
MS No.: amt-2016-400
MS Type: Research Article
Iteration: Revision

We would like to thank you and both reviewers for the useful remarks and comments on the manuscript. We have modified the manuscript as suggested and clarified some statements, where necessary. We hope that the revised manuscript has improved in respect to the original paper. Please find our answers to the comments of both reviewers below.

Sincerely yours,

Frederik Tack (Frederik.tack@aeronomie.be)

**G. Kuhlmann (Referee #1):**

First of all we would like to thank the reviewer. We greatly appreciate his positive comments and the very useful and constructive remarks. As described below, we have modified the manuscript according to suggestions and provided clarifications where necessary. We hope that the revised manuscript has improved in respect to the original paper. Please find a rebuttal against each point below.

***Black, bold, italic: Referee's comments***
Black: Author's reply
Green: Modifications in the manuscript

Note that based on the comments, two new figures were added to the manuscript, i.e. Fig. 14 and 19. References to figures in the author's reply refer to the new figure numbers.

***The paper describes the retrieval of high resolution NO2 maps from the airborne imaging spectrometer APEX over three cities in Belgium (Antwerp, Brussels and Liege). The authors develop a new retrieval algorithm for APEX similar to the algorithm described by Popp et al. (2012) but with several improvements such as better characterization of the influence of spatial binning and reference NO2 VCD on the retrieval as well as a correction term for in-flight changes of spectral resolution ("resolution cross section"). They also estimate VCD errors, which were not included in the paper by Popp et al., and compared their measurements with car mobile-DOAS measurements. The authors conclude that an airborne imager can be used for studying the spatial distribution of NO2 in urban areas.***

***The authors present new ideas, tools and data and their conclusions are mostly substantial. The topic is well within the scope of AMT. The scientific methods are valid and mostly clearly described. Some revisions are required, in particular, in Sections 3.1 and 3.2 which contain some small inaccuracies and/or not fully understandable results (see special comments). The paper is generally well written and the trail of ideas is understandable, but additional copy-editing is necessary.***

***1)***

***The authors state that one objective is to access APEX's capabilities for mapping NO2 in a "cost-effective way" (p. 2, l. 13ff). However, the cost effectiveness of APEX is not discussed in the paper. Thus, I suggest revising the objective or adding a section/paragraph on the cost effectiveness of APEX.***

We agree that the statement "cost-effective" in the objectives is not well chosen and not further discussed in the manuscript (which would also be out of the scope of it). We also refer to and agree with comment 2 of referee #2 that costs of an airborne campaign can be relatively high. But this depends to what you compare and on the desired specifications of the final $NO_2$ product. As this is something which is open for debate and, as mentioned, out of the scope of this study, we prefer to remove it from the objectives.

***2)***

*The authors describe the data as "time series" (p. 7, l. 4f and Fig. 4), but since the spatial dimension should be more important than temporal dimensions, I think a better term would be "along-track profile" as used in Fig. 14.*

Indeed. This is corrected accordingly.

*3)*

*The description of the smoothing effect and conclusion from this analysis are difficult to understand (p. 7, l. 6ff and Fig. 5). The authors should consider showing the raw data used for the fits in Fig. 5 and mark the location of the x-axis of Fig. 5 in Fig. 14.*

We have reformulated the analysis and conclusion and we hope that the description of the smoothing effect is more clear with respect to the original version (we have also marked the plume profile on Fig. 15): "The smoothing effect can also be observed in Fig. 5. A horizontal profile of approximately 2500 m was taken perpendicular to the second major plume and is indicated by a red dotted line in Fig. 15. Then, a Gaussian model was fitted to the obtained profile $NO_2$ VCDs for the three different binning levels. A broadening of the plume can be observed for the higher binning levels. The width of the fitted Gaussians is expressed as FWHM and broadens from 1685, 2129 to 2331 m for a binning of $8^2$, $20^2$ and $50^2$ pixels, respectively."

*4)*

*Paragraphs 3 and 4 need to be revised, because the conclusions are based on a wrong premise. APEX has no absolute pressure stabilization but is kept at 200 hPa above ambient pressure (Kuhlmann et al. 2016). Thus, CW positions depend on the ambient pressure at flight altitude during data acquisition and spectral shifts can occur between different campaigns. I think this is the most likely explanation for the differences found by the authors between the three datasets such as the shift between Antwerp and Brussels which were not flown on the same day.*

We want to thank the reviewer to point this out: the pressure in the spectrometer is indeed not absolutely stable but kept at 200 hPa above the ambient pressure and indeed this could largely explain the instabilities of the spectral performance. We have carefully read the publication of Kuhlmann et al. (2016), where the impact of the environmental conditions, and in particular in-flight pressure changes, on the spectral performance is studied. We have completely revised and reformulated paragraphs 3 and 4 of Sect. 3.2 as follows:
"During operation, airborne instruments are typically exposed to changes of environmental conditions (changes in pressure, humidity and temperature, vibrations, mechanical stress, etc.) which can affect the instrument characteristics and degrade its spectral performance. Even though the optical unit of APEX is temperature stabilized and sealed in a nitrogen atmosphere, kept at 200 hPa above ambient pressure, deviations can occur between the nominal and effective in-flight spectral performance. This has been extensively investigated by D'Odorico et al. (2011) and Kuhlmann et al. (2016) and confirmed by our study. Nominal parameters of the spectral performance, determined by the laboratory calibration, are given in Table 3. Based on laboratory measurements, the CW spectral shift should be smaller than 0.2 nm according to Schaepman et al. (2015). However, based on a recent re-analysis, a larger uncertainty between 0.4 and 1 nm was observed in real spectra (Kuhlmann et al., 2016). As a prism dispersion element is used, the FWHM is a non-linear function and broadening with wavelength. The slit function is assumed to have a Gaussian shape, with a

nominal FWHM of approximately 1.5 nm at 490 nm, i.e. the center of the fitting window, for the nadir detector element.

The spectral calibration on the in-flight spectra points out bigger CW shifts than specified by the nominal calibration as well as a broadening of the ISRF (see Table 3). In Fig. 6 a) the FWHM and b) the spectral shift and its dependency on the across-track scanline pixel position (spectral smile), are plotted for 490 nm for different across-track scanlines and flightlines: the mean of 20 scanlines 1) at the start and 2) at the end of flightline eight of the Antwerp data set (day 105), 3) at the start of flightline three of the Liège data set (day 105), and 4) at the start of flightline one of the Brussels data set (day 181). A second order polynomial has been fitted to each calibration set. In Fig. 6, minor changes of the slit function are detected between observations acquired on the same flightline. A spectral shift of approximately 0.18 nm can be observed between Antwerp flightline eight and Liège flightline three, acquired during a single flight (day 105) but with a time interval of approximately 1 hour. Large spectral shifts up to 0.8 nm can be observed between the Antwerp (day 105) and Brussels (day 181) flights. This is largely attributed to changing environmental conditions, with pressure changes in particular (D'Odorico et al., 2012).  As mentioned before, the pressure of the nitrogen gas in the spectrometer is kept at 200 hPa above ambient pressure. Pressure changes, due to changing ambient pressure or inaccuracies of the pressure regulation, can affect the index of refraction of the nitrogen gas and, subsequently, the dispersion at the prism, resulting in spectral shifts (Kuhlmann et al., 2016). This time-dependent variability of the APEX slit function in operational conditions can be critical for the analysis of the spectra, as discussed in Sect. 4.2.

The range of wavelength shifts in the across-track direction, exhibiting the typical smile curvature (Richter et al., 2011) in Fig. 6.b, is approximately 0.4 nm, which is well within the given tolerance of 0.35 pixels (Schaepman et al., 2015). The FWHM ranges between 2.4 nm, for the outer detectors, and 3.3 nm, for the nadir detector, which is more than twice the nominal value and at the limits to detect subtle spectral variations. According to Kuhlmann et al. (2016), there is no significant sensitivity of the FWHM to pressure changes (in contrast to the spectral shifts). Based on an intensive analysis, the study of Kuhlmann et al. (2016) concludes that the larger in-flight FWHMs are likely explained by a combination of (1) not fully corrected CCD readout smear, resulting in a spectral smoothing, and (2) filling-in of the Fraunhofer lines.
"

Please note that based on the comment, changes were made in other sections as well:

-Sect. 2.1: "The sealed optical unit is enclosed by a thermo-regulated box in order to be temperature (19° C ± 1° C) stabilised, while pressure in the spectrometer is kept at 200 hPa above ambient pressure (dry nitrogen atmosphere with partial differential pressure control)."

-Sect. 4.2: "The in-flight slit function is affected by environmental conditions during operation of APEX, despite the fact that the instrument is sealed, pressure- and temperature-regulated. Fig. 6 shows that slight changes of the slit function occur between different flightlines of a single flight. Large spectral shifts can be observed between different flights/campaigns."

-Sect. 7 :"(Recently a new in-flight spectral calibration algorithm, based on a maximum a posteriori optimal estimation approach, is proposed by Kuhlmann et al. (2016), improving the quality of the fit.)".

-Figure 6: "In-flight spectral calibration: a) the spectral resolution (FWHM) and b) the spectral shift and its dependency on the across-track scanline pixel position (spectral smile), plotted for 490 nm, i.e. the middle of the analysis window, for different flightlines. A second order polynomial has been fitted to each calibration set."

*5)*

*Furthermore, the CW accuracy of 0.2 nm does not include the spectral smile, which is defined as the range of wavelength shifts in across-track direction. Schaepman et al. (2015) state the spectral smile is smaller than 0.35 px (px = nm smile per nm nominal FWHM) which is about 0.5 nm at 490 nm. This is in good agreement with the spectral smile (about 0.4 nm and not 0.8 nm) the authors show in Figure 6b.*

We want to clarify that we were discussing the spectral shift that can occur between flightlines of different flights (around 0.8 nm) and not the spectral smile. But it was formulated in a bad and confusing way (Sect 3.2, paragraph 4, l7). Therefore we suggest the following adaptations:

Sect 3.2 paragraph 3: "…Based on laboratory measurements, the CW spectral shift should be smaller than 0.2 nm according to Schaepman et al. (2015). However, based on a recent re-analysis, a larger uncertainty between 0.4 and 1 nm was observed in real spectra (Kuhlmann et al., 2016)….". We also specified in Table 3 that the spectral shift of "< 0.2 nm" is valid for a single flight.

Sect 3.2 paragraph 4: "…In Fig. 6, minor changes of the slit function are detected between observations acquired on the same flightline. A spectral shift of approximately 0.18 nm can be observed between Antwerp flightline eight and Liège flightline three, acquired during a single flight (day 105) but with a time interval of approximately 1 hour. Large spectral shifts up to 0.8 nm can be observed between the Antwerp (day 105) and Brussels (day 181) flights. This is largely attributed to changing environmental conditions, such as temperature and pressure changes (D'Odorico et al., 2012).  As mentioned before, the pressure of the nitrogen gas in the spectrometer is kept at 200 hPa above ambient pressure. Pressure changes can affect the index of refraction of the nitrogen gas and, subsequently, the dispersion at the prism, resulting in a spectral shift (Kuhlmann et al., 2016)…"

We shortly addressed as well the spectral smile in Sect 3.2 paragraph 5: "…The range of wavelength shifts in the across-track direction, exhibiting the typical smile curvature (Richter et al., 2011) in Fig. 6.b, is approximately 0.4 nm, which is well within the given tolerance of 0.35 pixels (Schaepman et al., 2015)…"

*6)*

*APEX's in-flight spectral calibration of is described by D'Odorico et al. (2011) and Kuhlmann et al. (2016). The authors should consider citing these papers:*

*D'Odorico et al.:Performance assessment of onboard and scene-based methods for Airborne Prism Experiment spectral characterization, doi: https://doi.org/10.1364/AO.50.004755, 2011.*

*Kuhlmann et al.: An Algorithm for In-Flight Spectral Calibration of Imaging Spectrometers, doi: https://doi.org/10.3390/rs8121017, 2016.*

Thank you, both references were added to the manuscript. Please note that the first publication was already implicitly cited in:  "D'Odorico, P.: Monitoring the spectral performance of the APEX imaging spectrometer for inter-calibration of satellite missions, Remote Sensing Laboratories, Department of Geography, University of Zurich, 2012", where it appears as a chapter.
The second paper is indeed very relevant. However, at the time of writing the manuscript under evaluation, we were not aware yet of this recent publication.
Several references to the respective papers are made in the manuscript, e.g. in Sect. 2.1, 3.2 and 7.

*7)*

*Since a "synthetic resolution cross section" (p. 9, l. 23) is not commonly used in DOAS fits, the authors should provide additional explanations or a reference.*

This cross-section is based on a linearisation of the dependence of spectral Frauhofer lines on the slit function (or instrumental line shape ISRF) width. In fact, it corresponds to the first derivative of solar reference with respect to the slit function width. This cross-section allows to compensate for the impact of small changes of the instrumental resolution. Please see also Danckaert et al. (2015) and Beirle et al. (2017).

We have changed the paragraph in the manuscript to make this more clear: "Beside relevant high-pass filtered trace gas laboratory cross-sections ($NO_2$ and $O_4$), a low-order polynomial term, a synthetic Ring spectrum and a synthetic resolution cross-section are fitted to the logarithm of the ratio of the observed spectrum, and a reference spectrum. They account for respectively (1) smooth broad-band variations (e.g. reflection at the Earth's surface) and Rayleigh and Mie scattering; (2) the Ring effect (Grainger and Ring, 1962), i.e. the filling-in of Fraunhofer lines by rotational Raman scattering on air molecules, and (3) the impact of small changes of the instrumental resolution. The synthetic resolution cross-section is based on a linearisation of the dependence of spectral Frauhofer lines on the slit function width and corresponds to the first derivative of solar reference with respect to the slit function width (Danckaert et al., 2015; Beirle et al., 2017)."

*8)*

*While the differential approach reduces the impact of systematic instrumental stabilities and Fraunhofer lines, it does not necessarily neutralize them (p. 10, l. 10)*

Indeed, we scaled down this statement to: "This differential approach largely reduces the impact of both systematic instrumental instabilities and the prominent Fraunhofer lines in the spectra, which blur out the much finer trace gas absorption structures."

*9)*

*The authors state that spatial resolution of the NO2 map is 60x80m2. However, for visualization the maps are spatially smoothed by a Savitzky-Golay filter (p. 14, l. 9f). The authors should add sentence explaining how much this additional smoothing reduces the effective resolution of the NO2 map*

Maybe it would be better to reformulate these sentences in the manuscript and be more specific, as the Savitzky-Golay filter is basically a low-pass filter, removing the high frequencies or the short-scale variability, while retaining the main signal. The "smoothing" that can occur depends on the "smoothness" of the original signal and size of the filter. In the absence for example of high frequencies, the original signal would stay the same after application of the filter. Thereby, the filter is not applying a downsampling or averaging/smoothing in strict sense and subsequently it is difficult to give a general smoothing factor. Moreover, we think that the natural scale of the observed $NO_2$ variations appears to be larger than the spatial resolution of the $NO_2$ maps (60 x 80 m$^2$), due to the "natural" smoothing effect as a result of the viewing geometry and elevation of the $NO_2$ layer. We have tried to be more specific in the manuscript :"For visualisation purposes, the retrieved $NO_2$ VCD map is convolved by a low-pass filter in the across-track direction, reducing the high frequencies or

short-scale variability. The applied Savitzky-Golay filter is based on a least-squares fitting of a second order polynomial over a span of five pixels (Savitzky and Golay, 1964; Schafer, 2011) ".

*10)*

***The DOAS fitting error is not a random error, because it also contains systematic errors, e.g. from the parametrization (p. 15, l. 5)***

The DOAS fitting error can indeed have a systematic component as was mentioned on p.6,l.25. We propose to write "the error on the DOAS fit" in the manuscript.

*11)*

***I think the discussion of the results need additional data for supporting the conclusions. The authors should consider providing a time series of NO2 concentrations for the three sites. These data would help to interpret the bias expected between adjacent flight lines (p. 17, l. 1f) and the difference between the cities (p. 18, l. 5ff). In particular, the lower NO2 levels in Brussels might be also explainable by synoptic differences and the diurnal NO2 cycle (Antwerp was flown 11 LT and Brussels at 15 LT). In addition, additional labels on maps would help to follow the explanations (Figs. 14, 16 and 17). An NO2 time series would also help interpreting the bias between APEX and mobile-DOAS measurements (p. 20, l. 12ff)***

We thank the reviewer for these comments in order to improve the interpretation of the results and the manuscript. We have added an additional plot, i.e. Fig. 14, providing the $NO_2$ VCD diurnal variation retrieved at the Uccle MAX-DOAS station on April 15 and June 30, 2015. The MAX-DOAS observations can assist the interpretation of the $NO_2$ VCD variability during the flights, as well as the effect of the seasonal cycle. A new paragraph (paragraph 2) is added to Sect. 5, discussing Fig. 14: "The $NO_2$ VCD diurnal variation retrieved at the Uccle MAX-DOAS station on the campaign days, i.e. 15 April and 30 June 2015, are plotted in Fig. 14. The blue and red column indicate the flight time on April 15 over Antwerp (8:06 - 9:30 UTC) and on June 30 over Brussels (12:43 - 14:04 UTC), respectively. Further details of the station are provided in Sect. 6.2 and its location is indicated by a green dot in Fig. 17. Unfortunately there is no MAX-DOAS station so far in the Antwerp area. On both flight days, the Uccle station was upwind of the Brussels city center, thus in a semi-polluted area. During the April 15 flight, the $NO_2$ VCDs range between 4 and 21 x $10^{15}$ molec cm$^{-2}$, between 80° SZA sunrise and sunset, and VCDs increase approximately by 2.5 x $10^{15}$ molec cm$^{-2}$ during flight time. On June 30, $NO_2$ VCDs range between 4 and 7 x $10^{15}$ molec cm$^{-2}$. There is a slight decrease of 1.5 x $10^{15}$ molec cm$^{-2}$ during the APEX flight."
Note that we are aware that for April 15, we observe the VCD diurnal variation over the Uccle station which is approximately 40 km away from Antwerp. However, this is the closest MAX-DOAS station available. Currently, we are continuing the research reported in this manuscript and are comparing the APEX columns with $NO_2$ surface concentrations from in-situ stations (and model data). Due to the intrinsic differences between columns and surface concentrations and the strong effect of local sources on the in-situ measurements, we prefer not to provide in-situ $NO_2$ surface concentrations here. The comparison (or column-to-surface conversion) requires further research and would be out of the scope of this manuscript.
We agree with the reviewer that time of flight is also one of the factors explaining the lower $NO_2$ levels and we have adapted paragraph 6 accordingly. Note that we refer back to Fig. 14 in paragraph 6 in order to better explain the $NO_2$ seasonal cycle at the Uccle station: "The $NO_2$ VCD distribution map for the Brussels data set is shown in Fig. 17. The study area, consisting of the Brussels (sub)urban area and surrounding background, is covered by 95833 binned APEX pixels in approximately 80

minutes. Again a strong gradient can be observed, consistent with the southeasterly wind direction, with low $NO_2$ VCDs above the Sonian forest in the southeast and increased levels downwind of Brussels city and the ring road R0. The $NO_2$ VCDs in the Brussels data set are on average 7.7 ± 2.1 x $10^{15}$ molec $cm^{-2}$, with minima and maxima of 1 and 20 x $10^{15}$ molec $cm^{-2}$, respectively. In general, the $NO_2$ levels are almost 55 % lower than for the Antwerp data set. This is due to a combination of (1) the lack of significant industrial sources in the Brussels area, (2) the time of flight with respect to the traffic rush hours, and (3) the seasonality of $NO_2$ which tend to show maxima in winter and early spring. The $NO_2$ VCD diurnal variation retrieved at the Uccle MAX-DOAS station (Fig. 14) shows overall larger columns for the flight on 15 April, when compared to the flight on 30 June, with a mean $NO_2$ VCD of 10 and 5 x $10^{15}$ molec $cm^{-2}$, respectively, between 80° SZA sunrise and sunset. The seasonal $NO_2$ cycle, observed at the Uccle station, is discussed more in detail in Blechschmidt et al. (2017)."

In order to further improve the interpretation of the results, we have added additional labels in Fig. 15. Indeed it was difficult to understand to which stacks we were referring when discussing the main plumes. We have added the labels A-D to the plot in order to clarify this. Note that we were advised by the environmental agency to avoid giving the names of the industrial companies in the plot. In sect. 5, several small changes are made linked to the new labels: "…The main central plume with large extent is a double plume with main emissions from two chimney stacks at site "A", emitting respectively 30 and 95 kg of NOx per hour at an altitude of 70 m AGL, and a third stack northeast of it at site "B", emitting 145 kg of NOx per hour at an altitude of 35 m AGL…", "…There is actually a cluster of 30 to 40 stacks at site "A" and "B" which are contributing as well to the central plume.…"; Also a short description of the plumes at site "C" and "D" is added now to the manuscript: "…Smaller but clearly confined plumes origin from sites C and D, which are more isolated.…".

The main highways, contributing to the observed traffic emissions in the southeast, are better marked and labeled in Fig. 15, in a similar way as was done in Fig. 17. Also the plume cross-section, plotted in Fig. 5, was added to the plot.

In order to improve the interpretation to the reader, labels were also added to the plots in Fig. 18.

*12)*

***The authors might consider adding the route of the mobile measurements in Figures 14 and 16. The results for Liege are not shown and only briefly discussed in the paper with the reasoning that the results are similar to the other findings (p. 17, l. 19ff). However, since the Liege dataset is mentioned throughout the paper, I think the NO2 map for Liege should be added to the paper or the supplement for completeness.***

We prefer not to add the routes of the mobile measurements to the maps, which contain already much of information: when adding the routes, we get a strong overlap with the black vectorised ringroads and major highways, seriously reducing the clarity of the plots. Especially for Brussels, the vectorised roads (and junctions) are needed to interpret the results and understand the source of the main patterns of enhanced $NO_2$. We hope that the referee can agree with this choice. However, in order to further improve the interpretation of the mobile-DOAS measurements, we added additional information to the scatter plots in Fig. 21 by color-coding the points based on the absolute time offset between APEX and mobile-DOAS observations.

It's right that the Liège data set is mentioned throughout the manuscript, so indeed we should provide the map as well even if we keep the discussion short due to the stated reasons. We have added the $NO_2$ VCD map for Liège in Fig. 19. Please see also the reply to comment 1 of referee #2.

*13)*

*The authors should consider splitting Table 5 in two tables, because the headings for the last three rows (NO2 profile) seem not to match with the table content. In result, the quite long caption would be shorter, as well.*

The observation of the reviewer is correct and it is indeed a much better idea to split Table 5 in two. We have moved the $NO_2$ profile sensitivity study part to Table 6. The latter is restructured and the captions are rewritten.

*14)*

*I think the last column is probably showing the mean or median error for all VCDs and not the error of the total VCD.*

Indeed, this is not clearly formulated well in the original manuscript. We changed to "Column five shows the mean error for all retrieved VCDs." in the caption of Table 7 (this used to be Table 6!). We also changed the column header from "Total VCD" to "All VCDs".

*15)*

*p. 4, l. 18: "bulk" -> "majority"?*

Corrected. However "bulk of $NO_2$" or "$NO_2$ bulk" is also often used in the literature.

*16)*

*p. 7, l. 1f: What is the column co-located with?*

We understand the confusion and it is a bit difficult to formulate as we cannot say that we plot and compare the same column, due to the different binning levels. What we want to say is that the compared columns for the different binning levels are centered at the same along-track profile. We hope the following formulation is more clear: "Flightline eight of the Antwerp data set is binned according to three different binning levels, i.e. $8^2$, $20^2$ and $50^2$ pixels, resulting in an effective GSD of 24 x 32, 60 x 80, and 150 x 200 $m^2$, respectively. Then, the same north-south along-track profile is taken from the three different sets of retrieved VCDs."

*17)*

*p. 13, l. 3: "compensation factor" -> "enhancement factor" or "air mass factor"*

Corrected. "Air mass factor" is used.

*18)*

*p. 17, l. 20: 11u30 -> 11:30*

Corrected.

**19)**

*p. 20, l. 24: It is not clear what is meant by: "averaged by 2 subsequent retrievals".*

Because 1) the mobile measurements can show more variation than the airborne data due to very local emissions and 2) we want to reduce that one APEX pixel is sampled by two or more mobile measurements due to low speed, we average each two consecutive mobile measurements. We hope that the following formulation is more clear: "In order to reduce the impact of very local emissions and sampling of the same APEX pixel, the mobile data is averaged in bins of two consecutive measurements. "

**20)**

*p. 20, l. 29: "Gerrit Kuhlman" -> "Gerrit Kuhlmann"*

Corrected.

**21)**

*Table 1: I would suggest using "pixels" instead of "detectors" in the second row.*

Corrected. We applied a similar correction in Table 3.

**22)**

*Table 2: "Date" -> "Date (day of year)" and "0-180" -> "0, 180" in row flight pattern*

Corrected.

**23)**

*Table 4: Is "slope" a polynomial of degree 1?*

The intensity offset correction is indeed a polynomial of degree 1. In Table 4, we have replaced "slope" by "Polynomial order 1" in order to make this more clear.

**24)**

*Figure 6: "performed at" -> "shown for"?*

"Shown for" is indeed a more suitable formulation.

**25)**

***Figure 15/18: X-axis label: "Dec. time" -> "decimal time"?***

Yes, it is written fully in the revised version of the manuscript.

**Anonymous Referee #2:**

We greatly appreciate the positive feedback from the referee and the constructive comments. As described below, we have modified the manuscript according to suggestions and provided clarifications where necessary. We hope that the revised manuscript has improved in respect to the original paper. Please find a rebuttal against each point below.

*Black, bold, italic: Referee's comments*
Black: Author's reply
Green: Modifications in the manuscript

Note that based on the comments, two new figures were added to the manuscript, i.e. Fig. 14 and 19. References to figures in the author's reply refer to the new figure numbers.

*The paper by Tack et al., describes mapping of NO2 over Belgian cities using the APEX airborne spectral imager, an instrument that was not initially designed for the retrieval of trace gases. The authors present and their retrieval algorithm and discuss instrument performance during measurement flights. Airborne NO2 results are compared with collocated car observations and yield good agreement.*

*Overall the paper is well written and arguments are mostly easy to follow. The manuscript presents a good summary of the data analysis and NO2 results fulfills the in the introduction stated objective to assess the technical and operational capabilities of APEX NO2 mapping. However, a more detailed analysis of the instrument in-flight performance, as well as a discussion on possible improvements is missing, which would be very fitting for the scope of AMT (see also specific comments).*

*1)*

*Abstract: Please state right away that Liege results are not further discussed or leave out.*

We prefer to only mention Brussels and Antwerp in the abstract: "APEX data have been acquired under clear sky conditions over the two largest and most heavily polluted Belgian cities, i.e. Antwerp and Brussels on 15 April and 30 June 2015". However we mention the three sites in the introduction, but mentioning as well that the study focuses on Brussels and Antwerp: "In the framework of this study, dedicated APEX flights were performed above three of the largest and most heavily polluted urban areas in Belgium, namely Antwerp, Brussels and Liège. We report on the $NO_2$ retrieval scheme developed at BIRA-IASB and on the results of the flight campaigns performed in Belgium, with the focus on the cities of Antwerp and Brussels". Please note that we have also included the $NO_2$ map retrieved over Liège (Fig. 19) in Section 5 (see comment 12 of referee #1), however we keep the description short because of the mentioned reasons: "In the Liège data set, the highest $NO_2$ emissions are observed in the northeast, in the industrialised area of Herstal. The $NO_2$ levels range between 1 and 32 x $10^{15}$ molec cm$^{-2}$, with a mean VCD of 13.3 ± 3.1 x $10^{15}$ molec cm$^{-2}$. The overall error on the retrieved $NO_2$ VCDs is on average 23%".

*2)*

***Section 1: Typically airborne experiments are cost intensive field measurements compared to e.g. car measurements. Please explain why APEX measurements are described as "cost effective".***

We fully agree that costs of an airborne campaign can be relatively high, of course depending to what you compare. We also agree that the statement "cost-effective" is not well-chosen here (please see also comment 1 of referee #1). Our intension was to say that we want to retrieve high-resolution $NO_2$ maps in an efficient way. For example, we could interpolate $NO_2$ maps from many car-DOAS measurements in a city but this wouldn't be efficient/cost-effective. As this statement is open for debate and out of the scope of this study, we prefer to remove it from the objectives.

***3)***

***Section 2.1: p.4, line 21: Please define typical altitude, speed, integration time.***

This is corrected and reformulated as: "The ground sample distance (GSD) is 3 (across-track) by 4 (along-track) $m^2$ assuming a typical altitude of 6.1 km AGL, ground speed of 72 m.s$^{-1}$ and integration time of 58 ms,…"

***4)***

***Section 2.2: p.5, line 28: should say full NO2 columns below the plane.***

Corrected.

***5)***

***Section 3.1: p.6, line 14: How is the 58ms integration time chosen? Are there problems with saturated scans at times?***

This is the typical integration time for APEX flights in spectral unbinned mode. It was not specifically chosen for the $NO_2$ inference application, but it was chosen several years ago, after various tests were made by the APEX operators. It was concluded that an integration time of 58 ms ensures the best balance between the signal-to-noise ratio and the occurrence of saturated pixels.  We clarified this in the manuscript (not in Sect. 3.1 but at the first occurrence of the integration time in sect. 2.2 → see comment 3):" The ground sample distance (GSD) is 3 (across-track) by 4 (along-track) $m^2$ assuming a typical altitude of 6.1 km AGL, ground speed of 72 m·s$^{-1}$ and integration time of 58 ms, the latter being a good balance between the obtained signal-to-noise ratio and the occurrence of saturated scans."

***6)***

***Section 3.1: p.6, line 21: It is not clear which setting are used for the noise analysis, since Sect. 3.2 talks about calibration and analysis.***

Thank you for this observation. This reference is indeed wrong and should refer to section 4.1 instead of section 3.2. We reformulated this as: "The applied DOAS settings for the noise analysis are provided in Sect. 4.1 and Table 4"

*7)*

*Section 3.2: p.8: The strong variability of the instrument slit function is indeed a challenge for the DOAS analysis. Here further details on the instrument would be interesting, e.g., are there time traces of the instrument's pressure and temperature? Could these be correlated with the behavior of the slit function? Or even used to improve data analysis? The authors have optimized the DOAS analysis given the current state of the spectra, but recommendations of technical instrument improvements (if possible) or looking into exploiting technical in-situ data (if available) are missing, which would be well suited for the AMT audience.*

We thank the reviewer for this pertinent remark. The instability of the spectral performance is indeed one of the limitations and a big challenge for the DOAS analysis. Changing environmental conditions, with pressure in particular, are to our opinion the most likely explanation. The APEX laboratory and in-flight spectral performance was intensively investigated during recent years by RSL (Remote Sensing Laboratories - University of Zürich) and EMPA (Swiss Federal Laboratories for Materials Science and Technology) and results are reported in a number of publications (e.g. D'Odorico et al., 2011; D'Odorico et al., 2012; Hueni et al., 2014; Kuhlmann et al., 2015; Kuhlmann et al., 2016).

The PhD dissertation of D'Odorico (2012), focusing on the APEX spectral performance, states in the conclusion: "APEX performance measured during laboratory characterisation cannot be assumed for the operational environment. Causes of deviations are to be sought in the airborne operational environment, the most significant of which are pressure and temperature excursions." This was also mentioned in the manuscript on p.7, starting from ll.27. In the conclusion of D'Odorico (2012) recommendations for technical instrument improvements were made and applied: "…These findings led to an instrument revision aimed at the stabilization of the system for a range of temperature and pressure conditions to be encountered during operation. The revision included the manufacturing of a pressure regulation mechanism for the automatic release and fill-in of nitrogen according to the change in flight altitude. However, experiments carried out in the following year (2010), revealed that the implemented design revisions did not fully solve the pressure/temperature dependency of the system. Spectral shifts in the range of one spectral pixel were yet again estimated in flight for both, VNIR and SWIR detector." The APEX spectral performance and related instabilities are also investigated in the very recent work of Kuhlmann et al. (2016). Also a new in-flight spectral calibration algorithm, based on a maximum a posteriori optimal estimation approach, is proposed in order to improve the quality of the fit.

We would like to refer to the related comments 4, 5 and 6 of reviewer #1. Based on these comments and comments 7 and 14 of reviewer #2, Sect. 3.2 has been completely revised and reformulated. We hope that the causes of the spectral shifts and increased in-flight FWHM are better explained now and that the aforementioned publications, focusing on the instrument calibration and spectral performance, have a larger visibility.

Technical changes to the APEX instrument to improve trace gas retrieval are not evident due to several reasons: as mentioned in the manuscript, the APEX instrument has been primarily designed for environmental remote sensing of the land surface and APEX is almost exclusively operated for this purpose. Nevertheless, it has been demonstrated that atmospheric trace gases, such as $NO_2$, can also be retrieved with APEX. This study has revealed some limitations related to the spectral performance in order to detect the fine trace gas absorption structures. Remote sensing of the land surface is, however, much less constraining and the in-flight FWHM and spectral shifts are well within the tolerances for these kind of applications. Most of the time, acquisitions are performed in spectrally binned mode (groups of spectral bands are binned in the VNIR spectral region) in order to increase SNR, while retaining the high spatial resolution of 3 by 4 m.

The BUMBA, AROMAT and AROMAPEX projects have increased the knowledge and experience for trace gas retrieval based on hyperspectral imagers and brought together several groups and communities. Now the official APEX lifetime is shortening, discussions are started for a possible follow-up instrument, taking into account the findings of the related studies. If the follow-up instruments needs to be suitable for both LULC applications and atmospheric research, the current limitations should be tackled, e.g. by designing a vacuum enclosure.

D'Odorico et al.: Performance assessment of onboard and scene-based methods for Airborne Prism Experiment spectral characterization, doi: https://doi.org/10.1364/AO.50.004755, 2011.

D'Odorico, P.: Monitoring the spectral performance of the APEX imaging spectrometer for inter-calibration of satellite missions, Remote Sensing Laboratories, Department of Geography, University of Zurich, 2012.

A. Hueni et al.: "APEX: Radiometry under spectral shift conditions," *IEEE Geoscience and Remote Sensing Symposium*, Quebec City, QC, 2014, pp. 1381-1384. doi: 10.1109/IGARSS.2014.6946692, 2014.

Kuhlmann et al.: In-flight Spectral Calibration of the APEX Imaging Spectrometer using Fraunhofer Lines, ESA ATMOS Workshop, Heraklion, Greece, 2015

Kuhlmann et al.: An Algorithm for In-Flight Spectral Calibration of Imaging Spectrometers, doi: https://doi.org/10.3390/rs8121017, 2016.

**8)**

***Section 4.3.1: Is there in-situ temperature and pressure data available or ground measurements that could be interpolated? Please discuss choice of US Standard Atmosphere.***

Indeed, the used pressure and temperature profile is not correctly described in the manuscript, as "US Standard Atmosphere" should be "AFGL standard atmosphere for mid-latitude summer (Anderson et al., 1986)." This atmospheric profile is commonly used in AMF calculations and provides a good approximation. According to the study of Boersma et al. (2011), the effect of using a different atmospheric profile on the AMF calculations is very small: "using a midlatitude winter atmosphere profile (p,T) instead of a summer profile would change the tropospheric AMFs by 1 %." At the surface, the profiles have a different p of 5 hPa and a different T of 22 K.

Ground temperature and pressure are probably measured by the meteorological institute at different stations. However, the impact of including this in the AMF calculations would be very small.

Boersma, K. F., Eskes, H. J., Dirksen, R. J., van der A, R. J., Veefkind, J. P., Stammes, P., Huijnen, V., Kleipool, Q. L., Sneep, M., Claas, J., Leitão, J., Richter, A., Zhou, Y., and Brunner, D.: An improved tropospheric $NO_2$ column retrieval algorithm for the Ozone Monitoring Instrument, Atmos. Meas. Tech., 4, 1905-1928, doi:10.5194/amt-4-1905-2011, 2011.

**9)**

*Section 4.3.2: p.12, line 29: "similar statistics" here and elsewhere could go into an Appendix.*

We prefer to add the sensitivity study results for Brussels as well to Table 5 instead of in an appendix. We refer to it in Section 4.3.2 as: "Similar results were obtained for the Brussels data set and are provided in Table 5."

*10)*

*Section 4.3.2: p.13 and Section 4.6, p.15, line 27: Based on Fig. 16, the station in Uccle is in a semipolluted area, so any AOD measured there will not be representative for heavily polluted areas. The error on the aerosol effect seems underestimated. Please include an assessment using higher AODs.*

We thank the reviewer for pointing this out. We agree that the scarce data on the aerosol properties is a limitation and that indeed the Uccle station was in a semi-polluted area, taking into account the prevailing wind direction. During more recent flights over Berlin in the framework of the AROMAT/AROMAPEX campaign, we measured AODs in the same range but again the CIMEL station was outside the plume and thus in a semi-polluted area. Feedback on the campaign results were also focusing on more detailed data on the aerosol properties. This is something we should consider for future campaigns, e.g. by operating additional stations or a mobile system.

However we want to stress that flights were performed at clear-sky days with good visibility. We should also keep in mind that for example measurements in the plume (max. expected AODs) would also not be representative for the whole data set.

For the sensitivity tests, we assumed that all the aerosol extinction measured by the CIMEL takes place in the boundary layer, which leads to an upper estimate for the extinction in the BL, and thus the associated error. The difference of nadir AMF compared to a pure Rayleigh atmosphere is albedo-dependent and exhibits a maximum of 7%. This maximum estimated effect was further increased to 10% to account for higher aerosol loads We have clarified this in Sect. 4.3.2: "Radiative transfer simulations with a corresponding well-mixed extinction at the surface yield an albedo-dependent aerosol effect of 7% or less when compared to the AMFs computed based on a Rayleigh atmosphere. However, a relative uncertainty of 10% is considered for all flights, as the AERONET station in Uccle was in a semi-polluted area and furthermore to take into account the AOT variability".

*11)*

*Section 5, p.18, line 4: dSCDs around 0 are not "well above detection limit". Please be more specific with the respective statement (also in Fig. 15).*

This statement can be indeed confusing. The DSCDs depend on the, partly arbitrary, choice of the reference SCD. In the case there is much $NO_2$ in the reference, the DSCDs can be close to zero or even negative. We could state that the observed DSCD signal or DSCD variability is clearly popping out of the error bars/detection limit (1-sigma standard error), but maybe this is also confusing. Therefore we suggest to use "detection limit" only in the context of the VCDs. This is adapted in the text: "The 1-sigma slant error of APEX retrievals has a typical value between 3.4 and 4.4 x $10^{15}$ molec $cm^{-2}$ on the DSCD, corresponding to a detection limit of approximately 1.8 and 2.3 x $10^{15}$ molec $cm^{-2}$ on the VCD, assuming a typical AMF of 1.9 (See Sect. 4.3). In order for an absorber to be clearly identified, the retrieved column needs to be larger than this threshold." And in the caption of Fig. 16:

"The retrieved $NO_2$ VCDs are well above the detection limit of approximately 1.8 to 2.3 x $10^{15}$ molec $cm^{-2}$".

*12)*

*All Tables: Tables typically have a short caption. Please move explanations to text.*

It is often required that tables/figures and their captions are self-explanatory to the largest extent possible. Nevertheless, the caption of Table 5 was much too long and unclear. Therefore we splitted the original table in 2 parts: sensitivity study of 1) albedo/viewing/sun geometry (Table 5) and 2) a priori $NO_2$ profile (Table 6). Table 6 is restructured and the captions are reformulated. We would also like to refer to comment 13 of referee #1.

*13)*

*Table 1: Please choose detectors or pixel consistently. What does the plane speed relate to? The section "other" seems very random. I suggest integration in text where fitting.*

We thank the reviewer for these observations. "Detectors" is replaced by "pixels". Please see also comment 21 of referee #1, related to this. We replaced "plane speed" by "ground speed". The section "other" is indeed a bit random and is removed accordingly. However we kept "ground speed" and "integration time" in the table as this is strongly linked to the along-track spatial resolution.

*14)*

*Table 3: Can you comment on why the in-flight FWHM is significantly larger than nominal?*

This is a very pertinent but difficult question to answer. This was already observed in previous studies, e.g. Popp et al. (2012), without providing an explanation. D'Odorico (2012) states in the conclusion: "APEX performance measured during laboratory characterisation cannot be assumed for the operational environment. Causes of deviations are to be sought in the airborne operational environment, the most significant of which are pressure and temperature excursions."
In the very recent study of Kuhlmann et al. (2016), huge efforts were done to better understand the larger in-flight FWHMs. According to their study, there is no significant sensitivity of the FWHM to pressure changes (in contrast to the spectral shifts). The impact was studied of 1) filling-in of the Fraunhofer lines by the ring effect or underestimation of the $O_4$ absorption; 2) filling-in because of spatial binning of radiance spectra with slightly varying central wavelengths; 3) filling-in of dark lines because of radiometric calibration errors. However all of these effects seem too small to explain the larger in-flight FWHM. The authors conclude that probably a combination of not fully corrected CCD readout smear (resulting in spectral smoothing) and filling-in of the Fraunhofer lines causes the larger in-flight FWHMs. In sect. 3.2 we have added an explanation based on the analysis and conclusion of Kuhlmann et al. (2016):" Based on an intensive analysis, the study of Kuhlmann et al. (2016) concludes that the larger in-flight FWHMs are likely explained by a combination of (1) not fully corrected CCD readout smear, resulting in a spectral smoothing, and (2) filling-in of the Fraunhofer lines."

*15)*

***Table 4: Please explain "resol"***

This cross-section is based on a linearisation of the dependence of spectral Frauhofer lines on the slit function (or instrumental line shape ISRF) width. In fact, it corresponds to the first derivative of solar reference with respect to the slit function width. This cross-section allows to compensate for the impact of small changes of the instrumental resolution. Please see also Danckaert et al. (2015) and Beirle et al. (2017).

We have changed the paragraph in the manuscript to make this more clear: "Beside relevant high-pass filtered trace gas laboratory cross-sections ($NO_2$ and $O_4$), a low-order polynomial term, a synthetic Ring spectrum and a synthetic resolution cross-section are fitted to the logarithm of the ratio of the observed spectrum, and a reference spectrum. They account for respectively (1) smooth broad-band variations (e.g. reflection at the Earth's surface) and Rayleigh and Mie scattering; (2) the Ring effect (Grainger and Ring, 1962), i.e. the filling-in of Fraunhofer lines by rotational Raman scattering on air molecules, and (3) the impact of small changes of the instrumental resolution. The synthetic resolution cross-section is based on a linearisation of the dependence of spectral Frauhofer lines on the slit function width and corresponds to the first derivative of solar reference with respect to the slit function width (Danckaert et al., 2015; Beirle et al., 2017)."

**16)**

***Fig. 3: Is the reference noise taken into account here?***

For each binning level (i.e. the raw APEX spectra are spatially binned in the along- and across-track direction according to a power of 2, $2^n$, with $n$ ranging from 0 to 8), a clean reference area was binned in the same way. Making use of one "fixed" reference for all the different binning levels would be impossible because the reference spectrum should always be binned across-track in the same way as the spectra to be analysed. This way, all spectra from a "track" (including the reference spectrum) share the same optical configuration, taking into account that each detector is assumed to be a 1D-instrument with its own optical aberrations. This has been clarified in Sect. 3.1: "For a test area, the raw APEX spectra are spatially binned in the along- and across-track direction according to a power of 2, $2^n$, with $n$ ranging from 0 to 8. For each binning level, a clean reference area is binned in the same way".

**17)**

***Fig. 6: include description of lines in legend or caption.***

Ok, we have reformulated the caption as follows: "In-flight spectral calibration: **a)** the spectral resolution (FWHM) and **b)** the spectral shift and its dependency on the across-track scanline pixel position (spectral smile), plotted for 490 nm, i.e. the middle of the analysis window, for different flightlines. A second order polynomial has been fitted to each calibration set".

**18)**

***Please review manuscript for rules on spelling out digits.***

Thank you for noticing this. The whole manuscript was scanned to check and was corrected accordingly. The following rules were not always applied consequently:

"1) use words for cardinal numbers less than 10; use numerals for 10 and above (e.g. three flasks, seven trees, 6m, 9 days, 10 desks).
2) use numerals for cardinal number less than 10 for items that are unit of time or when used in scientific sense or mathematical sense."

**19)**

**p.3, line 1: Figs. 1 and 2**

Corrected. Also corrected on p.16, line 24; p.19, line 29 and p.20, line25

**20)**

**p.9, line 1 (SCDi) is already introduced**

Corrected.

**21)**

**p.9 Eqs. 2 and 3: "ref" should be subscript**

Corrected.

**22)**

**p.9, line 19: exchange word order of "wavelength" and "lower"**

Corrected.

**23)**

**p.11ff: Please be consistent with either Box-AMF or box-AMF.**

Corrected. Should be box-AMF.

**24)**

**p.15, end of line 22: "the" is missing**

I'm sorry but we don't find back where "the" is missing in one of these sentences.

**25)**

**p.17, line 20: 11:30 UTC**

Corrected.

**26)**

**Fig. 4: labels a) and b) are missing**

Corrected.

**27)**

**Fig. 6: Check rules on including units: "()" or "[]", cursive or not?**

We thank the reviewer for this observation. Square brackets should be used, as well as italic text. This was indeed not applied in a consistent way in the manuscript and is corrected accordingly in Fig. 6, 8, 16 and 20.

[revised manuscript text omitted]
. 4.  Flightline eight of the Antwerp data set is binned according to three different binning levels, i.e. $8^2$, $20^2$ and $50^2$ pixels, resulting in an effective GSD of 24 x 32, 60 x 80, and 150 x 200 $m^2$, respectively. Then, the same north-south along-track profile is taken from the three different sets of retrieved VCDs. As can be seen, the three $NO_2$ VCD along-track profiles show the same patterns of enhanced $NO_2$, consisting of two major plumes related to industrial activities in the Antwerp harbor. However, the $8^2$ binned data contains a lot of noise. On the other hand, the $50^2$ binned data smooths out effective $NO_2$ signals, e.g. at pixel 2050.  The smoothing effect can also be observed in Fig. 5. A horizontal profile of approximately 2500 m was taken perpendicular to the second major plume and is indicated by a red dotted line in Fig. 15. Then, a Gaussian model was fitted to the obtained profile $NO_2$ VCDs for the three different binning levels. A broadening of the plume can be observed for the higher binning levels. The width of the fitted Gaussians is expressed as FWHM and broadens from 1685, 2129 to 2331 m for a binning of $8^2$, $20^2$ and $50^2$ pixels, respectively.

**3.2 APEX spectral performance and wavelength calibration**

A key characteristic of the spectral performance is the instrument spectral response function (ISRF or slit function), being the response of the instrument to a signal as a function of wavelength. The ISRF can be determined by a peak response, i.e central wavelength (CW) and response shape, i.e. full width at half maximum (FWHM). Typically for pushbroom sensors, each across-track detector element should be considered as a 1-D instrument, due to optical aberrations and misalignments, with an intrinsic spectral response which is slightly different from the others.

In order to determine the spectral response of each pixel and to obtain a good alignment in the DOAS fit between the analysed spectra, reference and the absorption cross-sections, an in-flight spectral calibration is performed with the QDOAS software (Danckaert et al., 2015) prior to further processing. The accurate wavelength calibration aligns the Fraunhofer lines in the in-flight APEX spectra with a high resolution solar reference (Chance and Kurucz, 2010). The latter is iteratively

convoluted with an adjusted APEX slit function until a best match with the spectra is found, characterising the effective shift and FWHM. The APEX nominal wavelength-pixel relation, determined under laboratory conditions, is used as initial values in the calibration procedure in order to converge to the best solution. We determine FWHM and shift values in five sub-windows between 370 and 600 nm by minimizing the chi-square of the differences between the observed spectrum and the solar reference in each window. Then, we fit polynomials through the five resulting shift and FWHM values. As the pushbroom sensor consists of 1000 pixels across-track, which are binned by 20, the output consists of 50 different calibration sets in total.

During operation, airborne instruments are typically exposed to changes of environmental conditions (changes in pressure, humidity and temperature, vibrations, mechanical stress, etc.) which can affect the instrument characteristics and degrade its performance. Even though the optical unit of APEX is sealed in a nitrogen atmosphere and temperature stabilised, deviations can be significant between the nominal and effective in flight spectral performance. This has been extensively investigated in D'Odorico (2012) and confirmed by our study. Nominal parameters of the spectral performance, determined by the laboratory calibration, are given in Table 3. As a prism dispersion element is used, the FWHM is a non linear function and broadening with wavelength. The nominal FWHM has a Gaussian shape and is approximately 1.5 nm at 490 nm, i.e. the center of the fitting window, for the nadir detector element. Spectral shift of the CW from an absorption line should be smaller than 0.2 nm according to Schaepman et al. (2015).

The spectral calibration on the in flight spectra points out a broadening of the ISRF and bigger shifts than specified by the nominal calibration (see Table 3). In Fig. 6, the dependency of the FWHM and the shift on the scanline pixel position are plotted for 490 nm, and subsequently a second order polynomial has been fitted. The spectral calibration output is provided for several flightlines, i.e. 2 flightlines from the Antwerp data set (day 105), 1 flightline from the Liège data set (day 105) and 1 flightline from the Brussels data set (day 181). The FWHM ranges between 2.4 nm, for the outer detectors, and 3.3 nm, for the nadir detector, which is more than twice the nominal value and at the limits to detect subtle spectral variations. Also the spectral shift, exhibiting the typical smile curvature across the detector array (Richter et al., 2011), can be up to 4 times larger than the nominal value, e.g. 0.8 nm in case of the nadir detector on the Brussels flightline 10. Significant changes in FWHM and spectral shift can occur between different flightlines, largely attributed to changing pressure, temperature and environmental conditions, according to D'Odorico and Schaepman (2012) and Schaepman et al. (2015). In Fig. 6, minor changes of the slit function are detected between observations acquired on the same flightline. Bigger changes can, however, be observed between Antwerp flightline 8 and Liège flightline 3, acquired during the same flight (day 105) but with a time interval of approximately 1 hour. Substantial changes of both the FWHM and the spectral shift occur between the Antwerp (day 105) and Brussels (day 181) flight. This time dependent variability of the APEX slit function in operational conditions can be critical for the analysis of the spectra, as discussed in Sect. 4.2.

During operation, airborne instruments are typically exposed to changes of environmental conditions (changes in pressure, humidity and temperature, vibrations, mechanical stress, etc.) which can affect the instrument characteristics and degrade its spectral performance. Even though the optical unit of APEX is temperature stabilized and sealed in a nitrogen atmosphere, kept at 200 hPa above ambient pressure, deviations can occur between the nominal and effective in-flight spectral performance. This has been extensively investigated by D'Odorico et al. (2011) and Kuhlmann et al. (2016) and confirmed by our study. Nominal parameters of the spectral performance, determined by the laboratory calibration, are given in Table 3. Based on laboratory measurements, the CW spectral shift should be smaller than 0.2 nm according to Schaepman et al. (2015). However, based on a recent re-analysis, a larger uncertainty between 0.4 and 1 nm was observed in real spectra (Kuhlmann et al., 2016). As a prism dispersion element is used, the FWHM is a non-linear function and broadening with wavelength. The slit function is assumed to have a Gaussian shape, with a nominal FWHM of approximately 1.5 nm at 490 nm, i.e. the center of the fitting window, for the nadir detector element.

The spectral calibration on the in-flight spectra points out bigger CW shifts than specified by the nominal calibration as well as a broadening of the ISRF (see Table 3). In Fig. 6 a) the FWHM and b) the spectral shift and its dependency on the across-track scanline pixel position (spectral smile), are plotted for 490 nm for different across-track scanlines and flightlines: the mean of 20 scanlines 1) at the start and 2) at the end of flightline eight of the Antwerp data set (day 105), 3) at the start of flightline three of the Liège data set (day 105), and 4) at the start of flightline one of the Brussels data set (day 181). A second order polynomial has been fitted to each calibration set. In Fig. 6, minor changes of the slit function are detected between observations acquired on the same flightline. A spectral shift of approximately 0.18 nm can be observed between Antwerp flightline eight and Liège flightline three, acquired during a single flight (day 105) but with a time interval of approximately 1 hour. Large spectral shifts up to 0.8 nm can be observed between the Antwerp (day 105) and Brussels (day 181) flights. This is largely attributed to changing environmental conditions, with pressure changes in particular (D'Odorico et al., 2012). As mentioned before, the pressure of the nitrogen gas in the spectrometer is kept at 200 hPa above ambient pressure. Pressure changes, due to changing ambient pressure or inaccuracies of the pressure regulation, can affect the index of refraction of the nitrogen gas and, subsequently, the dispersion at the prism, resulting in spectral shifts (Kuhlmann et al., 2016). This time-dependent variability of the APEX slit function in operational conditions can be critical for the analysis of the spectra, as discussed in Sect. 4.2.

The range of wavelength shifts in the across-track direction, exhibiting the typical smile curvature (Richter et al., 2011) in Fig. 6.b, is approximately 0.4 nm, which is well within the given tolerance of 0.35 pixels (Schaepman et al., 2015). The FWHM ranges between 2.4 nm, for the outer detectors, and 3.3 nm, for the nadir detector, which is more than twice the nominal value and at the limits to detect subtle spectral variations. According to Kuhlmann et al. (2016), there is no significant sensitivity of the FWHM to pressure changes (in contrast to the spectral shifts). Based on an intensive analysis, the study of Kuhlmann et al. (2016) concludes that the larger in-flight FWHMs are likely explained by a combination of (1) not fully corrected CCD readout smear, resulting in a spectral smoothing, and (2) filling-in of the Fraunhofer lines.

[revised manuscript text omitted]

15   Beside relevant high-pass filtered trace gas laboratory cross-sections (NO$_2$ and O$_4$), a low-order polynomial term, a synthetic Ring spectrum and a synthetic resolution cross-section are fitted to the logarithm of the ratio of the observed spectrum, and a reference spectrum. They

20  account for respectively (1) smooth broad-band variations (e.g. reflection at the Earth's surface) and Rayleigh and Mie scattering; (2) the Ring effect (Grainger and Ring, 1962), i.e. the filling-in of Fraunhofer lines by rotational Raman scattering on air molecules, and (3) the impact of small changes of the instrumental resolution. The synthetic resolution cross-section is based on a linearisation of the dependence of spectral Frauhofer lines on the slit function width and corresponds to the first derivative of solar reference with respect to the slit function width (Danckaert et al., 2015; Beirle et al., 2017). O$_3$ and H$_2$O

[revised manuscript text omitted]

**4.4 Postprocessing: destriping**

After application of the retrieval equation, i.e. Eq. (3), a bias correction is applied on the retrieved $NO_2$ VCDs to cope with the across-track stripe-like pattern in the generated maps. Striping is inherent to pushbroom sensors due to the intrinsic spectral response of each detector which is slightly different from the others (see Sect.3.2). The applied bias correction is based on the algorithms presented in Boersma et al. (2011), Popp et al. (2012) and Lawrence et al. (2015). For each flightline, the column $NO_2$ values are averaged and a third order polynomial is fitted to the column averages. The deviation from the polynomial is treated as a detector dependent bias and used as a correction factor to subtract from the retrieved $NO_2$ columns. The procedure removes the across-track striping to a great extent, while retaining the $NO_2$ spatial patterns.  For visualisation purposes, the retrieved $NO_2$ VCD map is convolved by a low-pass filter in the across-track direction, reducing the high frequencies or short-scale variability. The applied Savitzky-Golay filter is based on a least-squares fitting of a second order polynomial over a span of five pixels (Savitzky and Golay, 1964; Schafer, 2011).

[revised manuscript text omitted]

The NO$_2$ VCD diurnal variation retrieved at the Uccle MAX-DOAS station on the campaign days, i.e. 15 April and 30 June

10    2015, are plotted in Fig. 14. The blue and red column indicate the flight time on April 15 over Antwerp (8:06 - 9:30 UTC) and on June 30 over Brussels (12:43 - 14:04 UTC), respectively. Further details of the station are provided in Sect. 6.2 and its location is indicated by a green dot in Fig. 17. Unfortunately there is no MAX-DOAS station so far in the Antwerp area. On both flight days, the Uccle station was upwind of the Brussels city center, thus in a semi-polluted area. During the April 15 flight, the NO$_2$ VCDs range between 4 and 21 x 10$^{15}$ molec cm$^{-2}$, between 80° SZA sunrise and sunset, and VCDs

15    increase approximately by 2.5 x 10$^{15}$ molec cm$^{-2}$ during flight time. On June 30, NO$_2$ VCDs range between 4 and 7 x 10$^{15}$ molec cm$^{-2}$. There is a slight decrease of 1.5 x 10$^{15}$ molec cm$^{-2}$ during the APEX flight.

In  Antwerp , the predominant anthropogenic emitters are mainly related to industrial activities in the harbor, but also to traffic in the southeast part. The port of Antwerp contains the biggest (petro)chemical cluster in Europe with branches of BASF, ExxonMobil, Solvay, Total, etc. The red dots in the NO$_2$ VCD distribution map (Fig. 15) represent

20    the most significant stacks, emitting more than 25 kg of NOx per hour, according to the emission inventory of the Belgian Interregional Environment Agency. The NO$_2$ field exhibits a strong gradient from west to east, consistent with the southwesterly wind direction. In the west, the NO$_2$ levels are low, around 3 to 7 x 10$^{15}$ molec cm$^{-2}$ due to the lack of major contributing sources. Substantial uncertainties can occur in this area as the levels are close to the detection limit. Downwind of the sources, the transported NO$_2$ is building up and patterns of enhanced NO$_2$ can be observed with maxima up to 3.5 x

25    10$^{16}$ molec cm$^{-2}$. The NO$_2$ VCDs are on average 1.7 ± 0.4 x 10$^{16}$ molec cm$^{-2}$. The detected plumes are clearly related to and transported from the most contributing stacks in the area. The main central plume with large extent is a double plume with main emissions from two chimney stacks at site "A", emitting respectively 30 and 95 kg of NOx per hour at an altitude of 70 m AGL, and a third stack northeast of it at site "B", emitting 145 kg of NOx per hour at an altitude of 35 m AGL. The plume is approximately 12 km long and is unfortunately not fully covered by the flightplan. There is actually a cluster of 30 to 40

30    stacks at site "A" and "B" which are contributing as well to the central plume. However, according to the emission inventory their emissions are mostly lower than 10 kg of NO$_x$ per hour. Smaller but clearly confined plumes origin from sites C and D, which are more isolated. Ship emissions are also expected to contribute to the observed NO$_2$ field, however these are hard to differentiate in this particular data set.

In the southeastern part of the NO$_2$ map (Fig. 15), NO$_2$ patterns can be observed, which are related to traffic emissions from the Antwerp city, the R1 ring road and the key highways E313 and E19. The two last flightlines are acquired around 11:00 - 11:30 AM LT and presumably the air masses containing the emissions from the rush hour are detected here, transported from the city and ring road R1 and building up, due to low wind speeds, northeast of the city. In the western part of the data set, an

[revised manuscript text omitted]

**Table 7.** Error budget analysis based on 73000 retrieved $NO_2$ VCDs of the Antwerp data set. Typical relative errors (percent) and absolute errors (x $10^{15}$ molec cm$^{-2}$ for $\sigma_{DSCD}$, $\sigma_{SCDref}$ and $\sigma_{VCD}$) are provided in column two to four, for small (< 33th percentile or < 1.4 x $10^{16}$ molec cm$^{-2}$), medium (33th to 66th percentile or 1.4 to 2.0 x $10^{16}$ molec cm$^{-2}$) and high (> 66th percentile or > 2.0 x $10^{16}$ molec cm$^{-2}$) $NO_2$ VCD retrievals, respectively. Column five shows the mean error for all retrieved VCDs.

| Error source | Small VCDs | Medium VCDs | High VCDs | All VCDs |
|---|---|---|---|---|
| $\sigma_{DSCD}$ | 40% (3.8) | 23% (3.9) | 15% (3.9) | 22% (3.9) |
| $\sigma_{SCDref}$ | 19% (1.8) | 11% (1.8) | 7% (1.8) | 10% (1.8) |
| $\sigma_{TAMF}$ | 15% (0.3) | 15% (0.3) | 15% (0.3) | 15% (0.3) |
| $\sigma_{VCD}$ | 29% (2.8) | 21% (3.5) | 18% (4.7) | 21% (3.7) |

[Figure]

**Figure 1.** OMI annual NO$_2$ VCD map for Belgium, 2015 and overview of the three Belgian cities where APEX flights were performed (Google, TerraMetrics, Giovanni, NASA GES DISC, OMNO2d, units in molec cm$^{-2}$).

[Figure]

**Figure 2.** Details of the APEX flightplan over Antwerp on 15 April 2015 and comparison with a spaceborne nadir OMI and TROPOMI pixel (Google, DigitalGlobe).

[Figure]

**Figure 3.** Allan plot illustrating the impact of spatial binning of the raw spectra on the RMS of the noise, plotted on a double logarithmic scale.

[Figure]

**Figure 4.** Along-track profile of **a)** NO$_2$ VCDs and **b)** RMS error of DOAS fits for 3 different levels of aggregated spectra, i.e. 8 by 8, 20 by 20, and 50 by 50 pixels. The VCDs are retrieved from an overlapping column on flightline eight of the Antwerp data set and are plotted from north to south.

[Figure]

**Figure 5.** Gaussian model fit to the $NO_2$ VCDs of a 2500 m horizontal profile, perpendicular to a major industrial $NO_2$ plume, for three different binning levels, i.e. $8^2$, $20^2$ and $50^2$ pixels, respectively.

[Figure]

**Figure 6.** In-flight spectral calibration: **a)** the spectral resolution (FWHM) and **b)** the spectral shift and its dependency on the across-track scanline pixel position (spectral smile), plotted for 490 nm, i.e. the middle of the analysis window, for different flightlines. A second order polynomial has been fitted to each calibration set.

[Figure]

**Figure 7.** Flowchart of the APEX NO$_2$ VCD retrieval algorithm.

[Figure]

**Figure 8.** Typical DOAS fit with **a)** red line, corresponding to the $NO_2$ molecular cross-section, convolved with the instrument slit function and scaled to the detected absorption in the measured spectrum (blue line) and **b)** the remaining residuals of the spectral fit.

[Figure]

**Figure 9. a)** APEX true color composite, **b)** APEX albedo level 2 product and **c)** computed TAMFs, for flightline eight of the Antwerp data set (15 April 2015). The strong dependency of the AMF to the albedo can be observed.

[Figure]

**Figure 10.** BAMF profiles illustrating the vertical sensitivity of the APEX instrument to $NO_2$. The high impact of the surface albedo, mainly on the lowest atmospheric layers, is shown based on five different scenarios, ranging from low to high albedo. Scenarios are based on the minimum, $\mu - 1\sigma$, mean, $\mu + 1\sigma$ and maximum albedo in the Antwerp data set.

[Figure]

**Figure 11.** Representative AURORA a priori NO$_2$ profiles, used for the Antwerp data set RTM calculations. A simple NO$_2$ box profile of 500 m height which is well-mixed in the boundary layer is used as well in the sensitivity study**.**

[Figure]

**Figure 12.** Scatter plot and linear regression analysis for the TAMF computation, based on two scenarios: integration of the BAMFs along (1) local $NO_2$ vertical profiles $A_{interp}$, interpolated on the AURORA model grid and (2) a fixed AURORA $NO_2$ profile $A_{harbor}$, with high $NO_2$ concentrations related to industrial sources.

[Figure]

**Figure 13.** Overall absolute (black dots) and relative errors (blue dots), $\sigma_{VCD}$, on the retrieved NO$_2$ VCDs, based on the Antwerp data set.

[Figure]

**Figure 14.** NO$_2$ VCD diurnal variation retrieved from the Uccle MAX-DOAS station on 15 April and 30 June 2015. The blue and red column indicate the flight time on April 15 over Antwerp and on June 30 over Brussels, respectively.

[Figure]

**Figure 15.** Retrieved NO$_2$ VCD field for Antwerp (15 April 2015) (Google, TerraMetrics). Red dots indicate the chimney stacks, emitting more than 25 kg of NOx per hour, according to the emission inventory of the Belgian Interregional Environment Agency. Four industrial sites, which are further discussed in Sect. 5, are indicated by the labels A-D. The red dotted line indicates the plume cross-section, plotted in Fig. 5. The blue vertical line indicates an along-track profile, for which the NO$_2$ DSCD and VCD time series are plotted in Fig. 16.

[Figure]

**Figure 16.** NO$_2$ DSCD and VCD time series for an along-track profile from north to south taken on flightline  of the Antwerp data set (15 April 2015). The retrieved NO$_2$  VCDs  are well above the detection limit of approximately 1.8 to 2.3 x 10$^{15}$ molec cm$^{-2}$. The profile, crossing the main plume from the harbor and the city center, is indicated by a dashed (blue) vertical line in Fig. 15.

[Figure]

**Figure 17.** Retrieved NO$_2$ VCD field for Brussels (30 June 2015) (Google, TerraMetrics). Blue squares indicate four NO$_2$ VCD hotspots, highlighted in Fig. 18. The green dot is the location of the Uccle MAX-DOAS station.

[Figure]

**Figure 18.** Enlargement of four peculiar NO$_2$ VCD hotspots in the Brussels data set (30 June 2015): **a)** major junction between the ring road R0 and the E19, and Brussels international airport; **b)** junction "place Meiser", close to the city center; **c)** eastern part of the E40 highway; **d)** gas turbine Drogenbos powerplant. The locations of the four zooms are indicated in Fig. 17 by blue squares (Google, TerraMetrics).

[Figure]

**Figure 19.** Retrieved NO$_2$ VCD field for Liège (15 April 2015) (Google, TerraMetrics).

[Figure]

**Figure 20.** APEX and Mobile-DOAS NO$_2$ VCD time series for **a)** the Antwerp flight (day 105) and **b)** the Brussels flight (day 181), respectively. The time offset between APEX and mobile-DOAS observations is plotted in dark grey. The NO$_2$ VCDs, measured by APEX (red dot) and the mini-MAX-DOAS (green dot) at the overpass over the Uccle station (13:21 UTC) are indicated on Fig. 20.b.

[Figure]

**Figure 21.** Scatter plot and linear regression analysis of the position-synchronised NO$_2$ VCDs, retrieved from APEX and mobile-DOAS for **a)** the Antwerp flight (day 105) and **b)** the Brussels flight (day 181), respectively. The points are color-coded based on the absolute time offset between APEX and mobile-DOAS observations.